# Unraveling gut microbiome alterations and metabolic signatures in hereditary transthyretin amyloidosis

Hanyu Li,[1] Zeyuan Wang,[1] Shan He,[1] Xinyue Zhao,[1] Qingyang Wu,[1] Yueshen Sun,[1] Yue Fan,[1] Xiaomin Hu,[2] Zhuang Tian,[1] Shuyang Zhang[1]

**ABSTRACT** Hereditary transthyretin amyloidosis (hATTR) is a rare, often fatal disease characterized by the abnormal aggregation of atypical transthyretin fibrils. Given the variability in the penetrance and clinical manifestations of hATTR, the role of nongenetic factors, particularly those related to the gut microbiota, warrants investigation. We conducted a cross-sectional study, examining the untargeted serum metabolome and gut metagenome in 13 patients with hATTR and 22 healthy controls. Significant disparities were observed in both the serum metabolome and gut microbiome of individuals with hATTR when compared to healthy controls. Notably, the serum levels of gamma-aminobutyric acid (GABA) and taurine were markedly decreased in the hATTR group, with the most pronounced reduction in those exhibiting hATTR-related cardiac amyloidosis. Additionally, commensals such as *Bifidobacterium pseudocatenulatum*, *Lactobacillus rogosae*, and *Hungatella hathewayi* were significantly diminished in hATTR patients and were positively correlated with the metabolite module containing GABA and taurine. Metagenomic and metabolomic pathway enrichment analyses collectively revealed disruptions in glutamate and taurine metabolism in hATTR. Our findings imply that patients with hATTR may exhibit metabolic irregularities in glutamate and taurine, potentially associated with an imbalance in the gut microbiota.

**IMPORTANCE** Hereditary transthyretin amyloidosis (hATTR) is influenced not only by genetic factors but also by environmental or host factors during its onset and progression. Previous studies have independently examined the metabolome or gut microbiome in hATTR, but the interplay between the microbiota and metabolism under this condition remains largely unknown. Our cross-sectional study represents the first comprehensive integration of gut metagenome and serum metabolome analyses in hATTR patients. We observed disturbances in glutamate and taurine metabolism among these patients, which correlated with distinctive shifts in the gut microbiota. This study offers insights into the intricate dynamics among gut dysbiosis, metabolic imbalances, and the progression of hATTR, suggesting directions for future research into the underlying mechanisms and therapeutic strategies.

**KEYWORDS** hereditary transthyretin amyloidosis, hATTR, cardiac amyloidosis, metabolome, metagenomics, microbiome

Transthyretin amyloidosis (ATTR) is a severe, progressive disorder characterized by neuropathy and cardiomyopathy (1), with an estimated prevalence of 1–9/100,000 (2, 3). The *TTR* gene on chromosome 18q12.1 encodes the transthyretin protein (TTR), a β-strand-rich tetramer prone to forming toxic amyloid fibrils (4). A single-point mutation within the *TTR* gene increases the risk of TTR misfolding and amyloid deposition, thereby leading to the hereditary form of ATTR (hATTR) (1, 5). The progression of hATTR is marked by gradual worsening of symptoms, increasing disability, and a decline in life expectancy,

Address correspondence to Xiaomin Hu, huxiaomin2015@163.com, Zhuang Tian, tianzhuangcn@sina.com, or Shuyang Zhang, shuyangzhang103@nrdrs.org.

Hanyu Li, Zeyuan Wang, and Shan He contributed equally to this article. The author order was determined by drawing straws.

The authors declare no conflict of interest.

See the funding table on p. 14.

with a median survival of 4.7 years after diagnosis (6). Apart from the age of onset and the specific *TTR* variant involved, the presence of cardiac amyloidosis (ATTR-CA) is a particularly concerning sign that points to a significantly poorer prognosis, with a median survival of 3.4 years (6, 7). Given the established relationships between distinct *TTR* variants and their associated phenotypes, which can vary from exclusive neuropathy to those overlapping with cardiomyopathy, the penetrance of *TTR* variants exhibits considerable variability. Furthermore, the targeted organ and disease progression show heterogeneity even among individuals with the same *TTR* variants (2). This inconsistency suggests that factors other than genetics also play a role in the development of hATTR.

The gut microbiota is pivotal in modulating human metabolism, the immune response, and neuroendocrine functions and is implicated in the development of various diseases (8, 9). Recent studies have explored the relationship between the gut microbiota and amyloidosis. A previous study demonstrated that the transfer of healthy microbiota in an animal model of Alzheimer's disease led to a decrease in amyloid β plaque formation (10). Additionally, a 16S rRNA sequencing study revealed notable disparities in the gut microbial composition between patients with immunoglobulin light-chain amyloidosis (AL) and healthy individuals (11). Despite these insights, research specifically examining the link between the gut microbiota and hATTR is lacking. Furthermore, the considerable influence of the gut microbiota on myocardial damage and neuropathy is gaining recognition (12–17). This has led to a question regarding the possible role of the gut microbiota in the affected organs in hATTR.

Recent studies have increasingly highlighted the role of microbiota-derived metabolites, such as short-chain fatty acids (SCFAs), secondary bile acids, and amino acid metabolites, in a spectrum of diseases (18–21). A study by Olsson et al. investigated the plasma metabolic profile of hATTR, identifying potential metabolic biomarkers for the early diagnosis of ATTRV30M (hATTR with the V30M *TTR* variant, which leads to a methionine for valine substitution at residue 30 of the mature TTR protein), such as xanthine and malic acid (22). However, the relationship between metabolism associated with the gut microbiota and the progression of hATTR has yet to be fully elucidated.

To address these questions, we performed a comprehensive approach involving metagenomic sequencing and untargeted metabolomics. The aims of this study were twofold: first, to identify distinctive alterations in the gut microbiota and serum metabolic profiles linked to hATTR; and second, to establish linkages between the gut microbiome, serum metabolome, and the clinical status of hATTR. Our findings suggested that individuals with hATTR might demonstrate metabolic dysregulations, particularly concerning glutamate and taurine, which could be associated with gut dysbiosis.

## RESULTS

### Serum metabolomic profiling in hATTR

The study enrolled 13 individuals with hATTR and 22 healthy controls (HCs; Table S1). The baseline characteristics of all hATTR participants are detailed in Table 1. The orthogonal partial least squares discriminant analysis (OPLS-DA) model revealed distinct serum metabolomic profiles between hATTR patients and healthy controls in both negative and positive ionization modes (see Fig. 1A and B for OPLS-DA scatter plots and permutation plots). A total of 118 metabolites were depleted, and 116 metabolites were enriched in hATTR, on the basis of a $P$-value < 0.05 and a variable importance in the projection (VIP) value >1 (refer to Fig. 1C for volcano plots).

Metabolite set enrichment analysis (MSEA) was conducted on the differentially abundant metabolites. As depicted in Fig. 1D, metabolites depleted in hATTR (left panel) were predominantly associated with amino acid metabolism, notably glutamine/glutamate, aspartate, and taurine. In contrast, metabolites enriched in hATTR (right panel) were significantly associated with energy metabolism pathways, such as the tricarboxylic acid (TCA) cycle and glyoxylate and dicarboxylate metabolism.

**TABLE 1** Summary of hATTR group[a]

| Patient no. | Sex | *TTR* variant | Age at onset | Age at diagnosis | Phenotype |
|---|---|---|---|---|---|
| 1 | F | p.Ala117Ser | 60 | 68 | Neurologic |
| 2 | F | p.Ala117Ser | 63 | 65 | Neurologic |
| 3 | M | p.Ala117Ser | 49 | 49 | Neurologic |
| 4 | M | p.Ala117Ser | 45 | 48 | Neurologic |
| 5 | M | p.Ala117Ser | 45 | 45 | Neurologic |
| 6 | M | p.Ser97Phe | 55 | 62 | Mix |
| 7 | M | p.Ser97Phe | 65 | 66 | Mix |
| 8 | M | p.Ser97Phe | 58 | 61 | Mix |
| 9 | M | p.Ser97Phe | // | 34 | Carrier |
| 10 | F | p.Val50Ala | 45 | 52 | Mix |
| 11 | F | p.Val50Ala | 53 | 56 | Mix |
| 12 | F | p.Val50Ala | 44 | 54 | Mix |
| 13 | M | p.Val50Ala | // | 33 | Carrier |

[a]F: female; M: male; neurologic: possessing clinical/instrumental evidence of peripheral and/or autonomic neuropathy in the absence of cardiac involvement; mix: neurologic phenotype and evidence of cardiac amyloidosis confirmed by electrocardiography and echocardiography; carrier: carrying the *TTR* mutation but exhibiting no evidence of organ injury; //: not available.

In brief, our findings highlight significant differences in serum metabolomic profiles between hATTR patients and healthy controls, suggesting potential disruptions in amino acid and energy metabolism that may be linked to hATTR.

## Major serum metabolites associated with cardiac amyloidosis in hATTR

We binned the differentially abundant metabolites into 95 modules (M1–M95) using Spearman correlations, as detailed in Table S2. A total of 20 modules were significantly correlated with the clinical phenotypes associated with cardiac involvement (Fig. 2A). Our analysis focused on seven modules that correlated with cardiac amyloidosis, given the substantial influence of this condition on the prognosis of hATTR patients. Specifically, modules M4, M47, and M62 were negatively correlated with cardiac amyloidosis. In contrast, modules M1, M10, M48, and M64 were positively correlated with cardiac amyloidosis. The differential levels of metabolites from key modules were compared between the hATTR-mixed, hATTR-neurologic, and HC subgroups, as depicted in Fig. 2B.

To assess the diagnostic potential of the selected metabolite modules, we generated receiver operating characteristic curves (ROC), as shown in Fig. 2C and Fig. S1A through D. Module M1, enriched in hATTR, demonstrated relatively good diagnostic performance with an area under the curve (AUC) of 0.897. Notably, compared with other metabolites in M1, 2-furoic acid (2-FA), potentially derived from the gut microbiota (23), showed relatively superior diagnostic potential, with an AUC of 0.897. Additionally, module M4, composed primarily of several amino acids and their derivatives (such as aspartic acid, gamma-aminobutyric acid [GABA], and taurine) which may be closely associated with the gut microbiota (24–26), exhibited a strong classification effect between the hATTR and HC groups, with an AUC of 0.941.

## Gut microbial changes associated with hATTR

We conducted metagenomic shotgun sequencing on fecal samples from all 35 participants to explore changes in the gut microbiota related to hATTR. The relative abundances of the dominant phyla in both the hATTR and HC groups are depicted in Fig. S2A. Firmicutes were predominant, with a relative abundance of 0.450 in HCs and 0.462 in hATTR patients. The microbial composition of the top 12 families in each sample is summarized in Fig. S2B.

Further analysis at the species level revealed that the hATTR group presented greater alpha diversity (Fig. 3A), as measured by the Shannon index ($P$-value = 0.0376, Wilcoxon rank-sum test) and Simpson index ($P$-value = 0.0077, Wilcoxon rank-sum test), than did

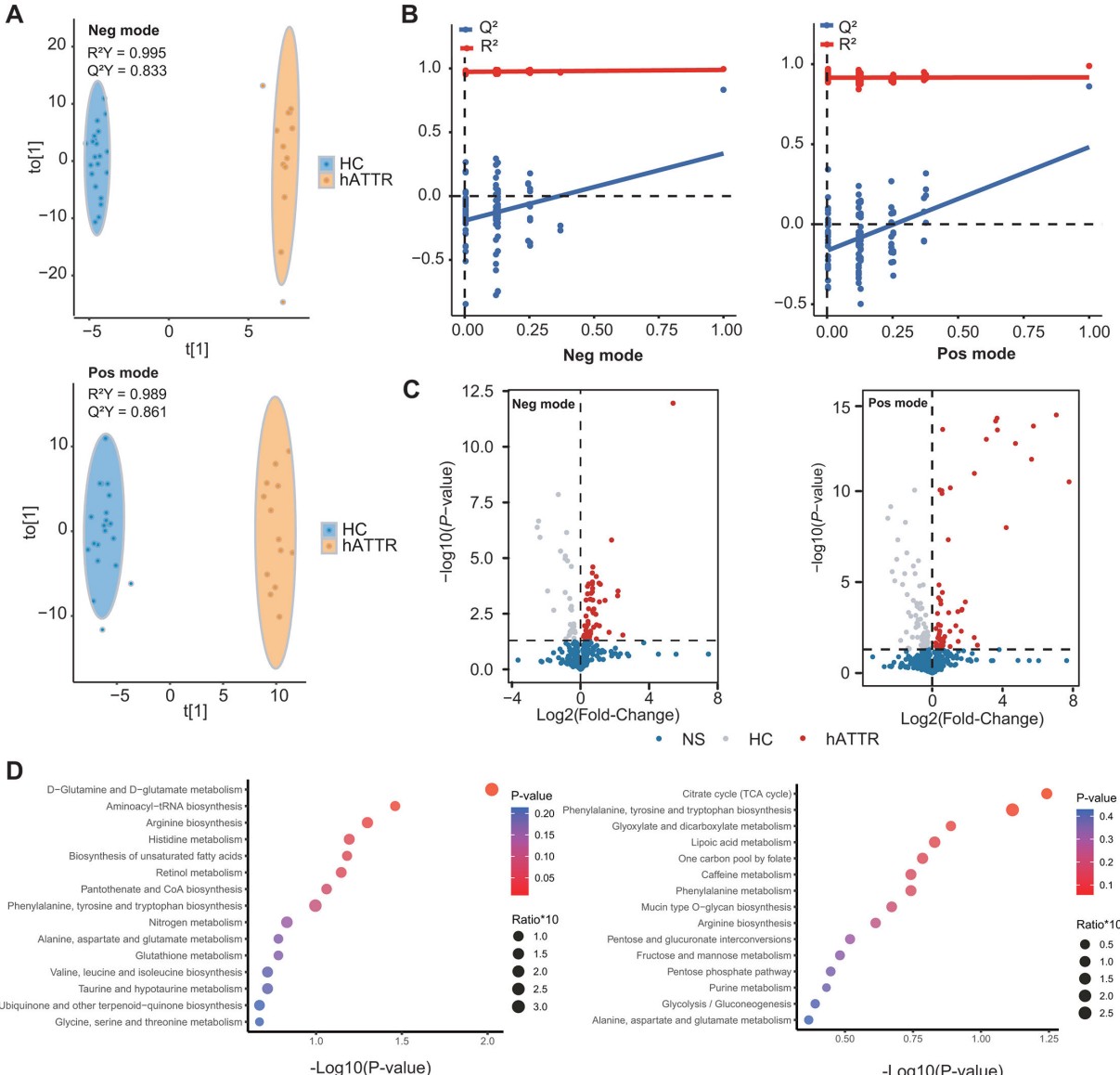

**FIG 1** Serum metabolomic alterations in hATTR patients. (A) OPLS-DA scores plots illustrating the separation between hATTR and HC samples in both negative and positive modes. (B) OPLS-DA permutation plots based on 99 permutations illustrating the validity of models in both negative and positive modes. (C) Volcano plots demonstrating differential serum metabolites between hATTR and HC groups. Data normalization was achieved by auto-scaling. Student's $t$-test. $P$-value < 0.05 is considered statistically significant. NS: not significant. (D) Visualization of metabolite set enrichment analysis (MSEA) based on hATTR-depleted metabolites (left panel) and hATTR-enriched metabolites (right panel). The "ratio" refers to the enrichment ratio of each metabolite set. Metabolites identified as hATTR-depleted or hATTR-enriched were selected based on the criteria of a Student's $t$-test $P$-value < 0.05 and an OPLS-DA VIP value >1.

the HC group. The partial least squares discriminant analysis (PLS-DA) model revealed a significant separation between the hATTR and HC groups at the species level ($R^2Y$ = 0.997 and $Q^2$ =0.574; Fig. 3B). Feature selection identified 56 species that were significantly enriched in hATTR patients and 50 species that were significantly enriched in HCs ($P$-value < 0.05, Wilcoxon rank-sum test; Fig. 3C).

By ranking them on the basis of PLS-DA VIP values, we found that species from SCFA-producing genera, such as *Faecalibacterium*, *Eubacterium*, and *Bifidobacterium*, were notably deficient in hATTR patients (Fig. 3D). Conversely, symbionts or pathogens such as *Bifidobacterium scardovii*, *Parabacteroides* sp. *Marseille-P3160*, and *Clostridiales bacterium CHKCl006* were significantly enriched in hATTR patients. We then performed a guild-based analysis to identify co-abundant species that might work as functional

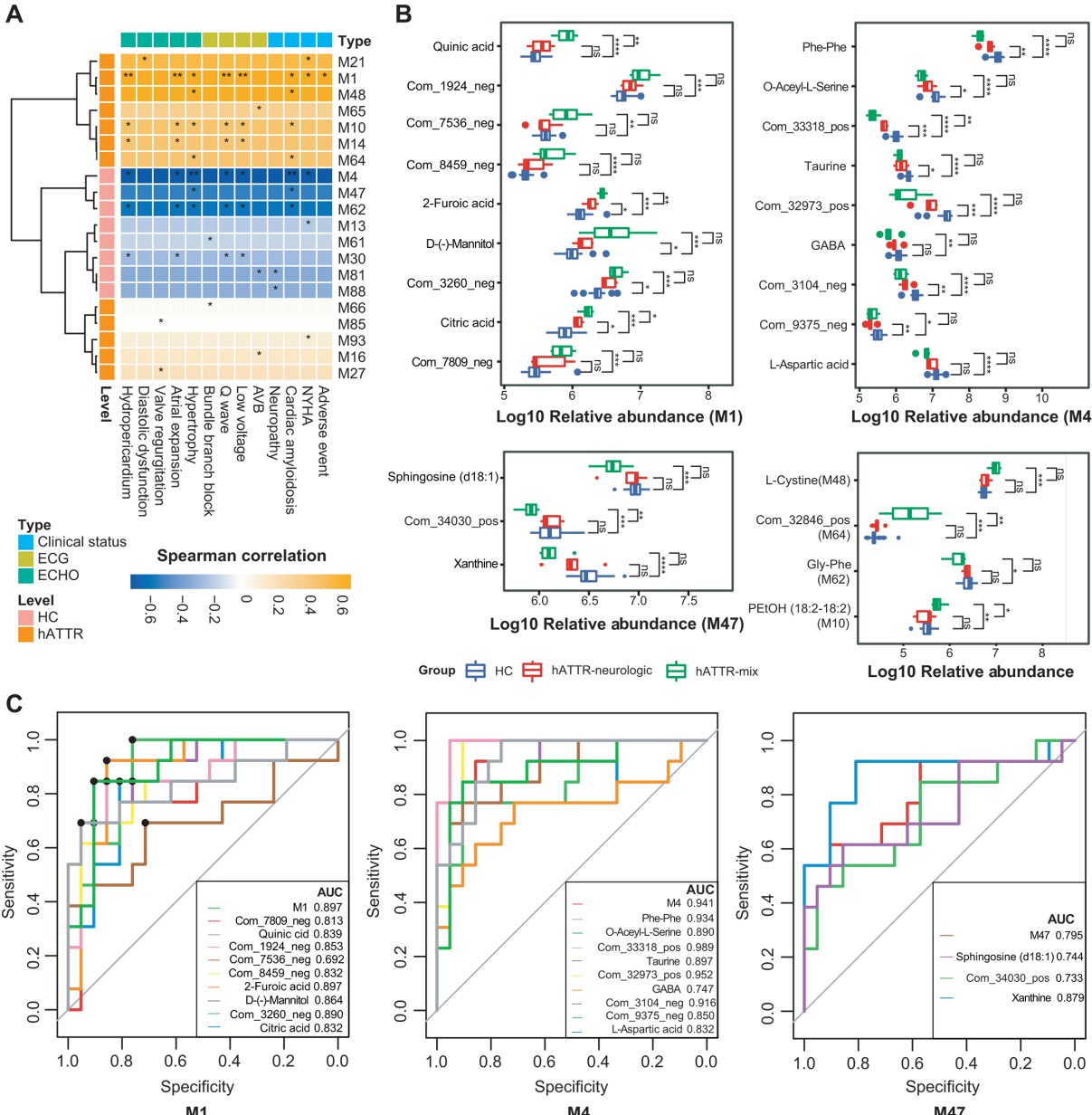

**FIG 2** Major serum metabolites associated with cardiac amyloidosis in hATTR. (A) The heatmap denotes Spearman's rank correlations between serum metabolite modules and clinical indices. *P-value < 0.05 and **P-value < 0.01. AVB = atrioventricular block. Adverse events were defined by history of syncope, sudden death, and/or malignant arrhythmia. (B) Serum metabolites belonging to modules correlated with cardiac amyloidosis. Wilcoxon rank sum test, *P-value < 0.05, **P-value < 0.01, and ***P-value < 0.0001. Boxes represent the interquartile ranges (IQRs), and the line inside the box represents the median; dots represent data points beyond the IQRs. (C) Receiver operating characteristic curves (ROCs) and area under the curve (AUC) of major serum modules. hATTR vs HC. Annotations of metabolites from different modules are detailed in Table S2.

groups in the gut ecosystem (27, 28). To achieve this, we calculated the Spearman correlations between the 106 key species and constructed a co-occurrence network (Fig. 3E). This analysis yielded a total of 46 co-abundant groups (CAGs; Table S3). The cumulative abundance of each CAG was correlated with metabolite modules and clinical parameters, as shown in the heatmaps in Fig. S3A and B.

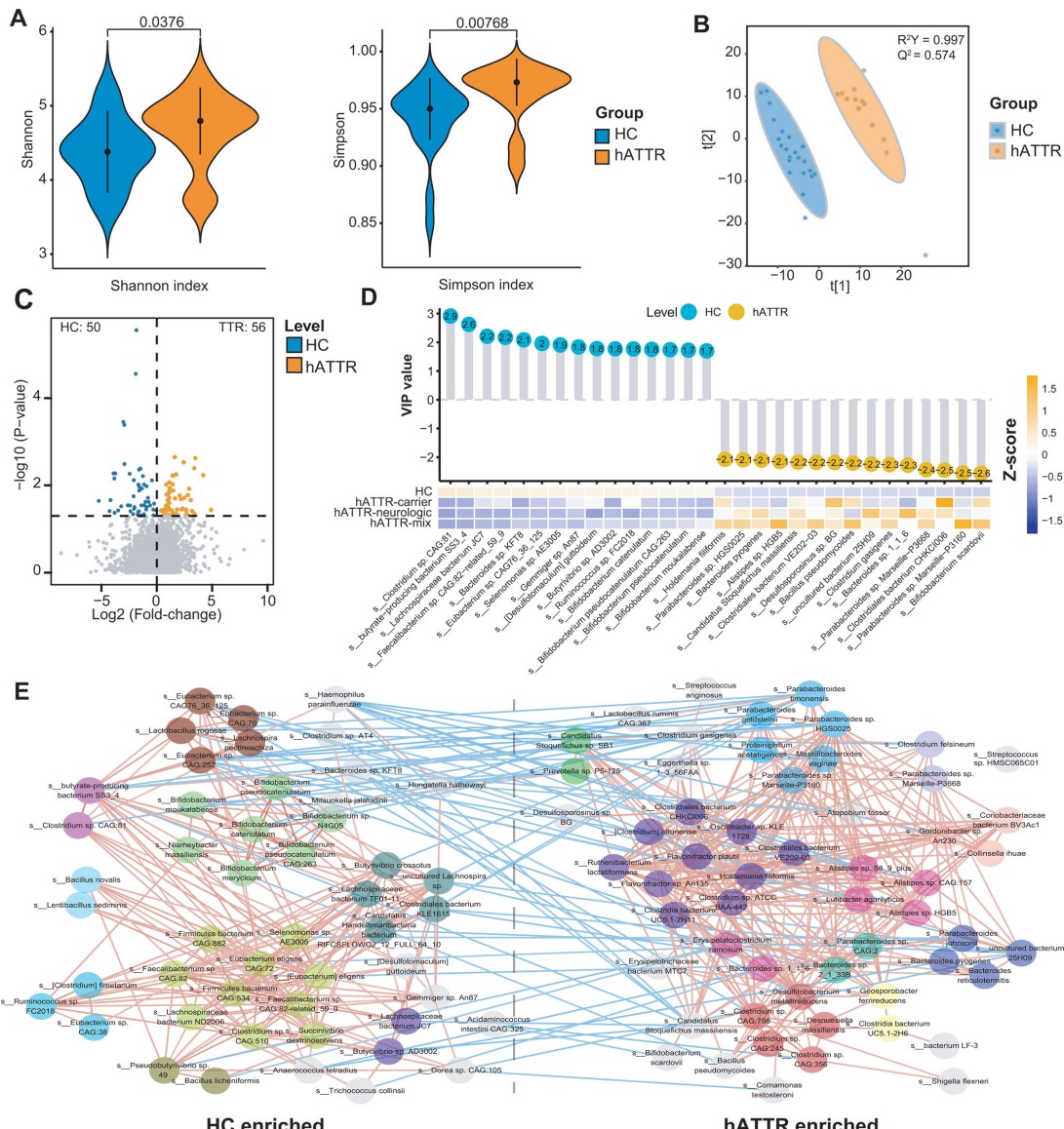

**FIG 3** Gut microbial alterations in hATTR. (A) Comparison of α-diversity based on species profiles between two groups. Left panel: Shannon index; right panel: Simpson index. Wilcoxon rank-sum test. (B) PLS-DA scores plot illustrating the separation between hATTR and HC samples based on species profiles. (C) Volcano plots demonstrating differential species between hATTR and HC groups. Wilcoxon rank sum test. *P*-value < 0.05 is considered statistically significant. (D) The lollipop chart (upper panel) denotes PLS-DA VIP scores of the top 15 hATTR-enriched and HC-enriched species. The heatmap denotes the differential enrichment of the 30 major differentially abundant species across subgroups. (E) Co-occurrence network deduced from differential species. Species are colored by co-abundant group (CAG). Except for *Streptococcus* sp. HMSC065C01 (CAG36), species not clustered with others are not shown in the figure. Red edges, Spearman's rank correlation coefficient >0.46, adjusted *P*-value < 0.05; blue edges, Spearman's rank correlation coefficient <−0.46, adjusted *P*-value < 0.05. Benjamini-Hochberg method was used for *P*-value adjustment.

## Multiomic analysis reveals the relationships between hATTR-associated microbiota and metabolites

By integrating the correlations between CAGs, metabolite modules, and clinical parameters, our multiomics analysis revealed potential microbiota-metabolism interactions associated with hATTR-related cardiac amyloidosis. As depicted in Fig. 4A, CAG10, composed of species from the *Parabacteroides* genus, and CAG36, dominated by *Streptococcus* sp. *HMSC065C01*, showed positive associations with cardiac involvement, mediated by metabolite modules M1, M48, and M64. Conversely, Fig. 4B illustrates that

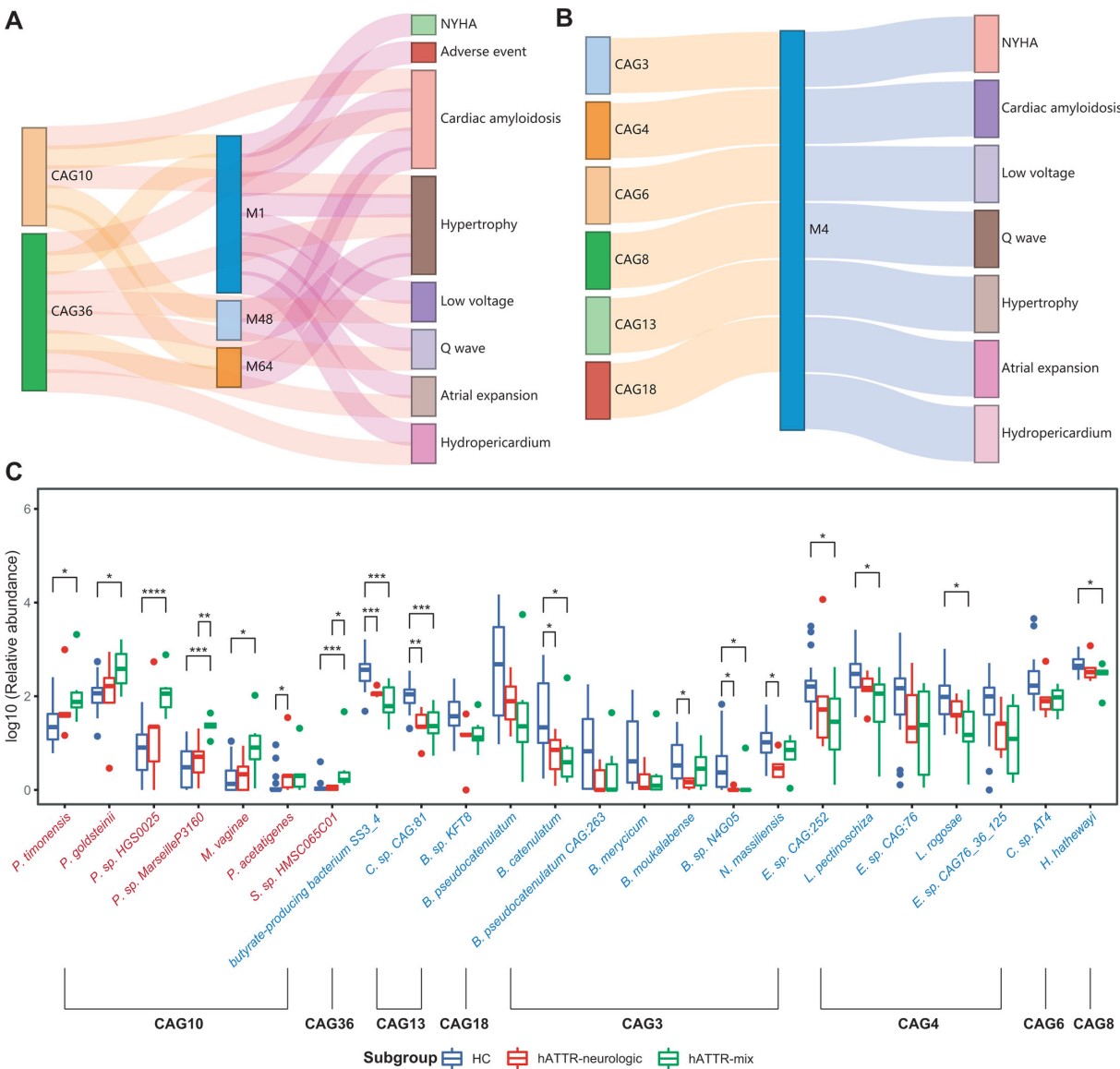

**FIG 4** Interrelationships between hATTR-associated microbiota, metabolites, and clinical indices. (A) The Sankey plot denotes positive correlations between hATTR-enriched CAGs, metabolite modules, and clinical indices characterizing hATTR-CA. (B) Orange connections indicate positive correlations between hATTR-depleted CAGs and M4. Blue connections indicate negative correlations between M4 and clinical indices characterizing hATTR-related cardiac amyloidosis. For (A) and (B), Spearman's rank correlation analysis was utilized. *P*-value < 0.05 is considered statistically significant. (C) Differential enrichment of species belonging to major CAGs across HC, hATTR-neurologic, and hATTR-mix subgroups. Wilcoxon rank sum test, \**P*-value < 0.05, \*\**P*-value < 0.01, and \*\*\**P*-value < 0.0001. Annotations of species from different CAGs were detailed in Table S3.

CAGs (CAGs 3, 4, 6, 13, and 18) rich in SCFA-producing species were negatively associated with cardiac involvement, with M4 playing a mediating role.

We observed that *Hungatella hathewayi* (CAG8), which was depleted in hATTR patients, exhibited a positive correlation with M4, a module that includes taurine. This finding is consistent with previous reports linking *H. hathewayi* to increased circulating taurine concentrations (29). Figure 4C presents the relative abundance of species from key CAGs across different subgroups, highlighting that species positively associated with the cardiac phenotype, particularly from CAG10 and CAG36, were enriched in the hATTR-mixed subgroup. In contrast, species negatively associated with the cardiac phenotype were depleted in the hATTR-mixed subgroup.

In brief, although whether the mentioned metabolites are directly produced by the key CAGs is yet to be determined, our multiomics analysis suggests that the gut microbiota might influence the levels of certain metabolic products, such as specific amino acids and their derivatives from module M4, thereby potentially impacting cardiac amyloidosis in hATTR patients.

## Distinct bacterial metabolic pathways in hATTR

To delve deeper into the metabolic capabilities of the microbiota associated with hATTR, we conducted a metagenomic functional analysis. Fig. 5A shows that carbohydrate metabolism, amino acid metabolism, and metabolism of cofactors and vitamins were the most prevalent among all the Kyoto Encyclopedia of Genes and Genomes (KEGG) metabolic categories across subjects. The volcano plot in Fig. 5B reveals that of the 5572 KEGG orthologs (KOs) analyzed, 213 KOs were significantly enriched in hATTR patients, whereas 150 were enriched in HCs.

As shown in Fig. 5C (left panel) and Table S4, KOs that were depleted in hATTR were predominantly associated with amino acid metabolism. Notably, a significant enrichment of KOs involved in alanine, aspartate, and glutamate metabolism was found, which aligns with our metabolome analysis. We found that aspartic acid, GABA, and some other glutamate derivatives, all of which are part of metabolite module M4, were depleted in hATTR. Furthermore, the KOs enriched in hATTR were enriched mainly in energy metabolism pathways, including glycolysis, pyruvate metabolism, and the TCA cycle, as depicted in Fig. 5C (right panel) and Table S5.

## DISCUSSION

The gut microbiota is recognized for its critical influence on human health, as it modulates both host metabolism and immune responses (16). However, the precise dynamics between hATTR and the gut microbiota remain elusive. In this study, we conducted an in-depth analysis of the gut microbiome and serum metabolome in individuals with hATTR. Our findings reveal disruptions within the gut microbiome of hATTR patients, characterized by an increase in opportunistic pathogens and a decrease in beneficial commensals. Additionally, we observed a distinctive pattern of metabolic dysregulation involving glutamate and taurine, which appears to be closely linked to the gut microbiota in the context of hATTR.

We confirmed that there are structural differences in the gut microbiota between hATTR patients and healthy controls. In terms of alpha diversity, hATTR patients presented a greater richness of gut species. The observed increase in microbial burden in hATTR could be linked to the deposition of amyloid proteins in the gut, a hypothesis that warrants further investigation through larger population studies and animal models for validation.

Among the microbes enriched in hATTR patients, *Bifidobacterium scardovii* exhibited the highest VIP value through PLS-DA analysis, as shown in Fig. 3D. It is a slow-growing, non-spore-forming anaerobic gram-positive bacillus. There are two reported cases of *Bifidobacterium scardovii* causing infections: one involving an elderly male patient with a breast abscess, and the other involving an elderly female patient with a history of cancer and chemotherapy who developed recurrent urinary tract infection (30, 31). Given the frailty of hATTR patients, whether the enrichment of *Bifidobacterium scardovii* is associated with a more fragile gut condition and a higher risk of infection requires further exploration. Our study also identified several species of *Parabacteroides* (within CAG10) that were positively correlated with cardiac amyloidosis (32). *Parabacteroides* has been found in relatively high proportions in individuals with gestational diabetes mellitus, hypertension, and chronic stress (33–37). The potential role of increased *Parabacteroides* spp. in the metabolic and neuropsychiatric aspects of hATTR warrants further exploration.

Regarding the hATTR-depleted microbes, CAG3 (featuring *Bifidobacterium pseudocatenulatum*) and CAG4 (comprising *Eubacterium* species and *Lactobacillus rogosae*) were

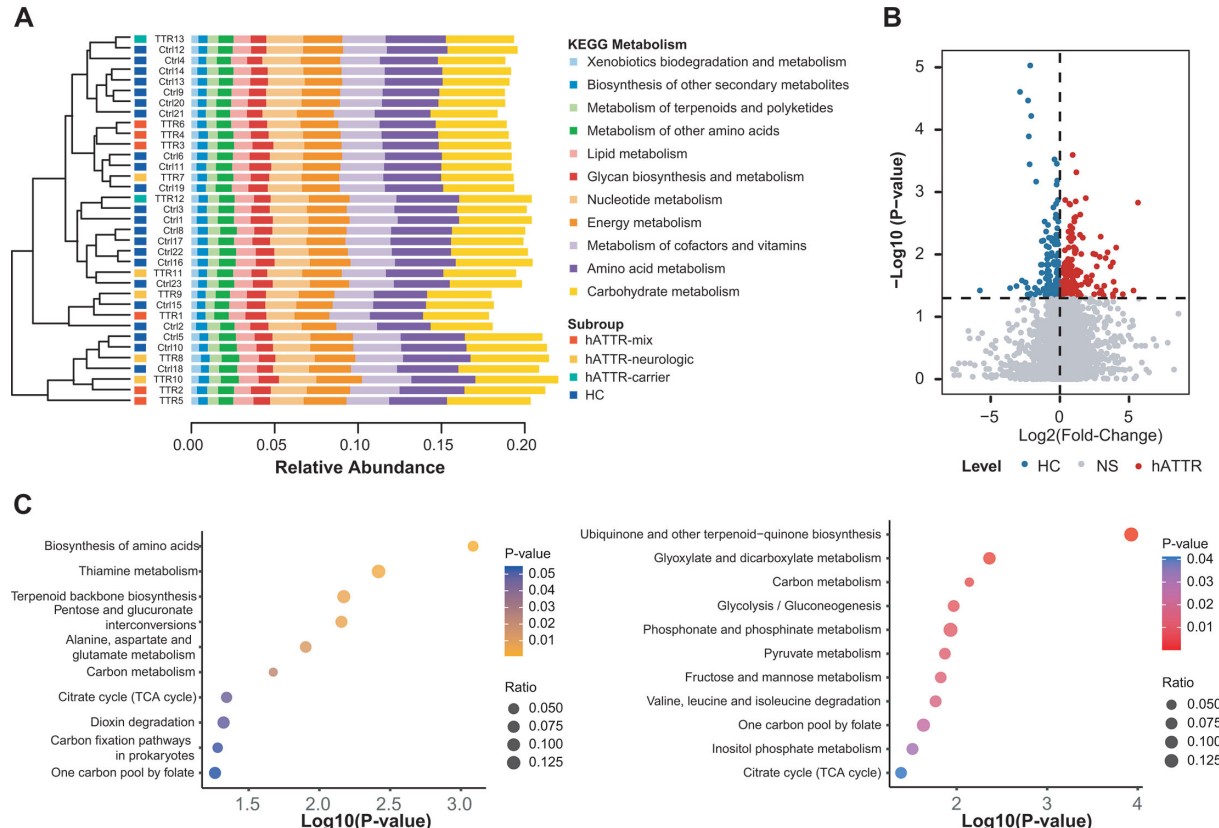

**FIG 5** Alterations of gut microbial metabolic pathways in hATTR. (A) Relative abundances of KEGG metabolisms across samples. The hierarchical clustering of samples was based on Bray-Curtis distance. (B) The volcano plot denotes differential KOs between hATTR and HC groups. Wilcoxon rank sum test. *P*-value < 0.05 is considered statistically significant. (C) Enriched KEGG metabolic pathways based on hATTR-depleted KOs (left panel) and hATTR-depleted KOs (right panel).

both found to be significantly downregulated in hATTR patients. It is important to note that *Bifidobacterium*, *Eubacterium*, and *Lactobacillus* are prominent genera known for their production of SCFAs (38, 39). SCFAs, particularly butyrate, are essential for maintaining intestinal barrier function and modulating mucosal immunity (40, 41). Whether the reduction of the above commensals leads to decreased levels of serum and fecal SCFAs needs to be further explored through targeted metabolomics. Additionally, whether the reduction of these commensals contributes to the disruption of the intestinal barrier in hATTR patients remains uncertain and requires further mechanistic studies.

An integrated analysis of the microbiome and metabolome revealed that multiple CAGs depleted in hATTR were positively correlated with metabolite module M4, which consists of various glutamate derivatives (Fig. 4B). M4 was downregulated in hATTR and negatively correlated with cardiac amyloidosis. Furthermore, metagenomic pathway analysis indicated that KOs downregulated in hATTR were enriched in pathways related to glutamate metabolism (Fig. 5C, left panel). Collectively, these findings suggest abnormalities in glutamate metabolism from both the metabolome and metagenome perspectives. Specifically, we observed downregulation of GABA in the hATTR group. GABA is a crucial inhibitory neurotransmitter that plays a role in numerous physiological and psychological processes. Previous studies have shown that GABA can be produced by various gut bacterial genera, such as *Bifidobacterium*, *Lactobacillus*, and *Bacteroides* (24, 42). Individuals with ATTR have been shown to be particularly susceptible to psychological distress and psychiatric disorders (43, 44). The relationship between this vulnerability and the dysregulation of glutamate metabolism warrants further investigation. GABA, which mediates the gut-brain axis, also has immunoregulatory functions,

promoting the differentiation of monocytes into anti-inflammatory macrophages that secrete interleukin-10 and suppress CD8+ T cell cytotoxicity (45). In our study, the results indicate that abnormal glutamate metabolism appears to be a distinctive feature of hATTR. The specific roles of GABA and other glutamate metabolites in this context require further exploration. Additionally, whether gut dysbiosis contributes to the disruption of glutamate metabolism in hATTR warrants further investigation.

We observed a positive correlation between hATTR-depleted *H. hathewayi* (CAG8) and metabolite module M4, which includes taurine. This correlation has been previously reported, with *H. hathewayi* increasing taurine levels in mice, potentially protecting against intracranial aneurysms (29). Taurine, a unique nonprotein amino acid with a sulfonic acid structure, is known for its cardioprotective effects, including anti-inflammatory and antihypertensive properties (46). Supplementation with taurine has also been demonstrated to increase exercise capacity in heart failure patients (46). Given the link between heightened inflammation and reduced survival rates in ATTR-CA patients (47), we hypothesize that *H. hathewayi* and taurine supplementation could benefit hATTR patients by mitigating inflammatory responses. Furthermore, taurine can form tauroursodeoxycholic acid (TUDCA) when it is conjugated with microbiota-derived ursodeoxycholic acid. TUDCA has been shown to disrupt transthyretin (TTR) fibril formation, particularly when combined with doxycycline (48, 49). The potential connection between disruptions in the bile acid pool and the progression of hATTR merits further investigation.

In alignment with a previous untargeted metabolomics study (22), our findings also revealed elevated levels of metabolites from the TCA cycle in the disease group, with citric acid being a notable example in our study. The TCA cycle plays a multifaceted role in physiological and pathological processes. Its dysregulation is linked to aberrant energy metabolism and oxidative stress, which can lead to various conditions, including cardiac dysfunction and acute stroke (50–53). The role of the TCA cycle in the pathogenesis of hATTR warrants further exploration. However, in contrast to the previous study, we observed a decrease in serum sphingosine (d18:1) and xanthine concentrations in hATTR patients. This discrepancy might stem from variations in genetic mutations and racial differences within the study populations. Given that metabolic profiles are influenced by a complex interplay of physiological and environmental factors, drawing identical conclusions across different studies is challenging.

Consistent with a prior investigation into the microbiota associated with hereditary transthyretin amyloidosis with polyneuropathy (54), our study also revealed increased alpha diversity in the gut microbiota of hATTR patients, with a notable predominance of Firmicutes over Bacteroidetes. A significant strength of Chen et al.'s research was their focus on hATTR patients harboring the p.Ala117Ser mutation, which likely minimized genotypic variability as a confounding factor. However, their reliance on the 16S V3–V4 region sequencing may have constrained the resolution of taxonomic classification.

To our knowledge, this study represents the first comprehensive integration of gut microbiome and serum metabolome analyses in hATTR. We have identified a range of differentially abundant metabolites and microbial species that exhibited significant associations with hATTR. Importantly, our multiomics approach revealed that disruptions in microbiota-associated glutamate and taurine metabolism may be distinctive features of hATTR. Given the rarity of hATTR, our study is limited by its limited sample size. This limitation may affect the robustness of the statistical results and the generalizability of our findings. Future studies should aim to validate our conclusions with larger hATTR cohorts. Furthermore, while this study included three distinct *TTR* genotypes, the potential impact of genotype variation on serum metabolomics and the gut microbiota composition remains unclear and could introduce bias. Another limitation is the lack of analysis of dietary factors, given the relatively well-established association between diet and the gut microbiota (55). The integration of dietary data and further exploration of the impact of dietary factors on hATTR are strongly suggested. Additionally, *in vivo*

animal models could provide valuable insights into the specific roles of the key microbial species and metabolites identified in the pathophysiology of hATTR.

In conclusion, our study revealed a distinct pattern of gut dysbiosis in hATTR, characterized by a reduction in commensals, such as *Bifidobacterium pseudocatenulatum*, *Lactobacillus rogosae*, and *H. hathewayi*, and an increase in symbionts or opportunistic pathogens, such as *Parabacteroides* spp. Additionally, we have identified perturbations in the metabolism of glutamate and taurine in hATTR. GABA and taurine may serve as key metabolites that bridge the gut microbiota with the pathophysiology of hATTR, warranting further investigation.

## MATERIALS AND METHODS

### Study population

In this cross-sectional case-control study, we enrolled individuals diagnosed with hATTR at Peking Union Medical College Hospital (PUMCH). The diagnosis was confirmed by genetic screening of *TTR* mutations, assessment of clinical symptoms, family history evaluation, echocardiography (UGC), and biopsy, as described in our previous study (4). The study cohort comprised patients with confirmed *TTR* gene mutations; participants with ATTRwt were not included. Any amyloidosis patients with elevated levels of light chains detected in the blood, urine, or bone biopsy were excluded.

The phenotypes were classified as follows: (i) clinically unaffected phenotype (hATTR-carrier): individuals who carry the *TTR* mutation but show no signs of organ damage. (ii) Neurologic phenotype (hATTR-neurologic): those with clinical or instrumental indications of peripheral and/or autonomic neuropathy without any cardiac involvement. (ii) Mixed phenotype (hATTR-mix): individuals presenting with both a neurologic phenotype and evidence of cardiac amyloidosis confirmed by electrocardiography (ECG) and UCG. ECG signs for cardiac amyloidosis included low voltage (electrocardiogram QRS amplitude ≤0.5 mV in all limb leads and/or <1 mV in all precordial leads), advanced atrioventricular block, and/or intraventricular conduction disturbances. UCG signs suggesting cardiac amyloidosis included increased ventricular wall thickness (the end-diastolic thickness of the interventricular septum was >1.2 cm), a granular sparking appearance of the ventricular myocardium, increased thickness of the atrioventricular valves or interatrial septum, and pericardial effusion (1). Any other possible causes of cardiac hypertrophy were excluded.

For *TTR* genotyping, as described in our previous publication (4), genomic DNA was isolated from whole peripheral blood. Exons 2, 3, and 4 of the *TTR* gene (GenBank: accession number m11844) were amplified by PCR (Takara ExTaq polymerase). The primers used for amplification are listed in Table S6. The amplified products were further sequenced via an ABI Prism 3130 automated sequencer (4).

Controls were defined as healthy volunteers without ATTR-related clinical signs, symptoms, or family histories. Subjects were excluded if they had organic gastrointestinal diseases, malignant tumors, autoimmune diseases, sepsis, or severe renal dysfunction (serum creatinine >3.0 mg/dL) or if they were administered antibiotics for more than 3 days in the previous 3 months.

For the recruited participants, fasting venous blood and fecal samples were collected. The detailed protocols for the preparation and storage of the serum and stool samples were described in the Supplementary methods. The collected metadata covered participants' demographic and anthropometric features. The New York Heart Association functional class was utilized to assess the activity tolerance of the participants (56). ECG, UGC, and laboratory data were collected at PUMCH. Adverse events in hATTR patients were defined by the history of syncope, sudden death, and/or malignant arrhythmia. This study was approved by the Ethics Committee of Peking Union Medical College Hospital (JS-1233) and was performed in accordance with the principles of the Declaration of Helsinki. All the subjects provided written informed consent to participate in the study.

## Untargeted metabolomic study

Serum metabolomics was conducted by ultra-high performance liquid-chromatography-MS/MS (UHPLC-MS/MS) using a Vanquish UHPLC system (ThermoFisher, Germany) coupled with an Orbitrap Q Exactive HF mass spectrometer (Thermo Fisher, Germany) at Novogene Co., Ltd. (Beijing, China). Only features with coefficient variation values <30% in the quality control samples were filtered for downstream analysis. Metabolites were annotated using the KEGG (https://www.genome.jp/kegg/pathway.html), HMDB (https://hmdb.ca/metabolites), and LIPIDMaps (http://www.lipidmaps.org/) databases. Statistical analysis was performed using the Student's *t*-test and OPLS-DA (SIMCA v14.1 from Umetrics, Sweden) to identify differentially abundant metabolites with a significance level of $P < 0.05$ and a VIP value greater than 1. Differentially abundant metabolites depleted or enriched in hATTR were separately uploaded to MetaboAnalyst 5.0 for MSEA (57).

We then performed a hierarchical clustering analysis of the differential metabolites using the R package vegan. First, the distance between differential metabolites was calculated via the Spearman method. Clusters, termed metabolite modules (labeled M1–M95), were subsequently identified via the hclust and cutree functions with a height threshold of 0.4 and the "average" method. The cumulative abundance of each module was then calculated by summing the abundances of individual metabolites within the module. The resulting abundance matrix of metabolite modules was used for subsequent multiomics statistical analyses.

## Metagenomic sequencing and processing

We performed shotgun metagenomics sequencing. The extraction of fecal DNA via the cetyl trimethyl ammonium bromide (CTAB) method and library preparation was accomplished at Novogene Bioinformatics Technology Co., Ltd (detailed in the Supplementary methods) (58). Illumina PE150 pair-end sequencing was conducted on the Illumina NovaSeq 6000 platform. Quality control of the raw sequencing data was performed using Readfq (V8, https://github.com/cjfields/readfq). Host-origin reads were filtered out using Bowtie2 version 2.2.4 (https://bowtie-bio.sourceforge.net/bowtie2/index.shtml) with the following parameters: --end-to-end, --sensitive, -I 200, and -X 400. The clean reads were assembled using SOAPdenovo software (V2.04; parameters: -d 1, -M 3, -R, -u, -F, and -K 55). Single-sample assembly was conducted. The acquired scaffolds were interrupted from the N connection to yield scaftigs (i.e., continuous sequences within the scaffolds). A mixed assembly was then conducted. Bowtie2.2.4 was applied to acquire reads that were unmapped to scaftigs in each sample (parameters: --end-to-end, --sensitive, -I 200, and -X 400). All unmapped reads were subsequently merged and assembled using SOAPdenovo (V2.04) with the same parameters. Only scaftigs longer than 500 bp were retained for downstream analysis. The MetaGeneMark (V2.10; http://topaz.gatech.edu/GeneMark/) software was used to predict open reading frames (ORFs). The predicted ORFs were then subjected to redundancy removal using the CD-HIT software (V4.5.8; http://www.bioinformatics.org/cd-hit; parameters: -c 0.95, -G 0, -aS 0.9, -g 1, and -d 0). The abundance of Unigenes in each sample was determined by counting the number of aligned reads (Bowtie2, parameters: --end-to-end, --sensitive, -I 200, and -X 400) and normalizing by gene length.

## Taxonomic and functional annotation and quantification

Taxonomic assignments for Unigenes were accomplished by aligning the sequences to the NR database using DIAMOND software (V0.9.9; https://github.com/bbuchfink/diamond/; parameters: blastp, -e 1e-5). The final taxonomic annotation for each Unigene was ascertained using the lowest common ancestor-based algorithm, which selects annotation with an *e*-value less than or equal to the minimum *e*-value * 10. The taxonomic abundance at any given level was determined by aggregating the abundances of all genes classified under that taxonomic group. Functional assignments for

Unigenes were achieved by blasting the sequences against the KEGG database (version 2018–01–01; http://www.kegg.jp/kegg/) via DIAMOND software with the parameter setting of blastp, -e 1e-5. The top Blast Hit for each sequence was selected for further analysis. The relative abundance of each KEGG functional category was calculated by summing the relative abundances of all genes annotated to that particular functional level.

## Microbiome analysis

To assess alpha diversity, we calculated the Shannon index and Simpson index via the Vegan package (v2.5–6) in R v4.0.2 on the basis of the species profiles. For feature selection, we conducted a multivariate statistical analysis using PLS-DA (SIMCA software v14.1, Umetrics, Sweden) to differentiate hATTR samples from controls. VIP values were calculated to assess the significance of the variables. The Wilcoxon rank-sum test was applied for significance evaluation at the species level. Species with a $P$-value < 0.05 and an average relative abundance $>10^{-6}$ were identified as key species.

A hierarchical clustering analysis of the key species was then performed using the R package vegan. The distances between key species were computed using the Spearman method via the vegan package. The co-occurrence network of key species was visualized using Cytoscape v3.8.0. Then, clusters, termed CAGs (labeled CAG1-46), were identified via the hclust and cutree functions with a height threshold of 0.5 and the "average" method. The cumulative abundance of each CAG was then calculated by summing the abundances of individual species within the group. The resulting abundance matrix of CAGs was used for subsequent multiomics statistical analyses.

For metagenomic functional analysis, we uploaded the abundance matrix of KOs to the MicrobiomeAnalyst 2.0 platform (https://www.microbiomeanalyst.ca/) (59). The cumulative abundance of each KEGG metabolism category was calculated, and the Wilcoxon rank-sum test was used to evaluate the significance of the KOs. Differential KOs with a $P$-value < 0.05 were then subjected to pathway enrichment analysis using MicrobiomeAnalyst 2.0.

## Multiomics correlation, statistics, and visualization

Spearman correlations among metabolite modules, CAGs, and clinical parameters were computed using SPSS software (v26.0). To depict the connections between significant metabolite modules, CAGs, and clinical parameters, we employed Sankey plots generated with the networkD3 R package. The bioinformatics visualizations were conducted in R version 4.0.2 and RStudio. We leveraged several R packages for diverse visualizations: ggplot2 and ggpubr for creating scatter plots, volcano plots, bubble plots, lollipop charts, violin plots, box plots, and bar plots; pheatmap for generating heatmaps; and pROC for plotting ROCs and calculating the AUC values of key metabolites.

## ACKNOWLEDGMENTS

We appreciate the support of Peking Union Medical College Hospital. We thank all the participants in this study.

This study was supported by the Beijing Nova Program, grant number 20220484031, the Chinese Natural Science Foundation, grant number 82270405, National High Level Hospital Clinical Research Funding, grant numbers 2022-PUMCH-A-026, 2022-PUMCH-B-016, and 2022-PUMCH-B-098; and the CAMS Innovation Fund for Medical Sciences, grant number CIFMS 2021-I2M-1-003.

## AUTHOR AFFILIATIONS

[1]Department of Cardiology, State Key Laboratory of Complex Severe and Rare Diseases, Peking Union Medical College Hospital, Chinese Academy of Medical Sciences & Peking Union Medical College, Beijing, China

²Department of Medical Research Center, State Key Laboratory of Complex Severe and Rare Diseases, Peking Union Medical College Hospital, Chinese Academy of Medical Sciences & Peking Union Medical College, Beijing, China

## AUTHOR ORCIDs

Xiaomin Hu http://orcid.org/0000-0003-4107-1692
Zhuang Tian http://orcid.org/0000-0003-1140-4062
Shuyang Zhang http://orcid.org/0000-0002-1532-0029

## FUNDING

| Funder | Grant(s) | Author(s) |
|---|---|---|
| Beijing Nova Program | 20220484031 | Xiaomin Hu |

## AUTHOR CONTRIBUTIONS

Hanyu Li, Conceptualization, Data curation, Formal analysis, Investigation, Methodology, Visualization, Writing – original draft | Zeyuan Wang, Data curation, Formal analysis, Investigation, Visualization, Writing – original draft | Shan He, Conceptualization, Formal analysis, Investigation, Writing – original draft | Xinyue Zhao, Formal analysis, Investigation, Validation | Qingyang Wu, Investigation, Validation, Writing – review and editing | Yueshen Sun, Investigation, Methodology, Validation, Writing – original draft | Yue Fan, Formal analysis, Investigation, Software | Xiaomin Hu, Conceptualization, Data curation, Formal analysis, Funding acquisition, Methodology, Resources, Supervision, Writing – review and editing | Zhuang Tian, Conceptualization, Investigation, Resources, Supervision, Writing – review and editing | Shuyang Zhang, Conceptualization, Funding acquisition, Methodology, Resources, Supervision, Writing – review and editing

## DATA AVAILABILITY

The metagenomic sequencing data supporting the results of this study have been deposited in the Genome Sequence Archive (GSA) for human under BioProject PRJCA015966, with the accession code HRA004297.

## ETHICS APPROVAL

The study was conducted in accordance with the Declaration of Helsinki and approved by the Ethics Committee of Peking Union Medical College Hospital (JS-1233). Informed consent was obtained from all the subjects involved in the study. Written informed consent has been obtained from patients to publish this paper.

## ADDITIONAL FILES

The following material is available online.

### Supplemental Material

**Supplemental figures (Spectrum02302-24-s0001.pdf).** Figures S1 to S3.
**Supplemental methods (Spectrum02302-24-s0002.docx).** Additional methods.
**Supplemental tables (Spectrum02302-24-s0003.xlsx).** Tables S1 to S6.

### Open Peer Review

**PEER REVIEW HISTORY (review-history.pdf).** An accounting of the reviewer comments and feedback.

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
