## [Reviewer comments · Microbiology Spectrum]

Microbiology Spectrum

Unraveling Gut Microbiome Alterations and Metabolic Signatures in Hereditary Transthyretin Amyloidosis

Hanyu Li, Zeyuan Wang, Shan He, Xinyue Zhao, Qingyang Wu, Yueshen Sun, Yue Fan, Xiaomin Hu, Zhuang Tian, and Shuyang Zhang

Corresponding Author(s): Xiaomin Hu, Peking Union Medical College Hospital

Review Timeline:

Submission Date:	September 11, 2024
Editorial Decision:	January 17, 2025
Revision Received:	March 25, 2025
Accepted:	April 2, 2025

Editor: Jan Claesen

Reviewer(s): The reviewers have opted to remain anonymous.

Transaction Report:

DOI: <https://doi.org/10.1128/spectrum.02302-24>

Re: Spectrum02302-24 (Unraveling Gut Microbiome Alterations and Metabolic Signatures in Hereditary Transthyretin Amyloidosis)

Dear Dr. Xiaomin Hu:

Thank you for the privilege of reviewing your work. Below you will find my comments, instructions from the Spectrum editorial office, and the reviewer comments.

Thanks for submitting your research to Spectrum and again my apologies for the delay in getting this decision back to you. Your work has now been evaluated by two independent Reviewers who are enthusiastic about the research (as am I). The Reviewers pointed out some comments and suggestions for improving the manuscript and I would be happy to consider a revised version that addresses these in a point-by-point manner.

Revision Guidelines

Sincerely,
Jan Claesen
Editor
Microbiology Spectrum

Reviewer #2 (Comments for the Author):

The study reports the first comprehensive integration of gut metagenome and serum metabolome analyses in patients with

hereditary transthyretin amyloidosis. This study emphasizes the dynamics between the gut microbiota and disease and further underscores the potential roles of biomarkers in understanding the underlying mechanisms of and therapeutic strategies against diseases. Given the significant impact of the study, I strongly suggest that the authors improve the overall flow of the manuscript, taking into consideration the following remarks:

1. In Line 39 (Importance), you mentioned that hATTR is influenced also by environmental or host factors. Is diet a factor of this disease? If so, is it also a factor that should be looked into to explore the complex crosstalk between the gut microbiome and disease? Are there previous studies that explored the correlation of diet with the disease?
2. In Line 87, the term "ATTRV30M" is not clear. Please define it.
3. What was the statistics used to determine the sample size (13 patients with hATTR and 22 healthy controls)? Does this sample size suffice for gut microbiome analyses?
4. In the Results, can you clarify the term "modules" used for serum metabolites?
5. The authors mentioned that the controls were age- and sex-matched with the patients. What was the inclusion criterion for age?
6. I recommend that the authors list down the primers used for the TTR gene.
7. I strongly suggest that the authors provide a brief methodology regarding blood and fecal sample collection (volume of blood, weight of fecal samples) and storage.
8. Please clarify in the methods that you performed shotgun metagenomics sequencing.
9. Cite the study protocol used for the CTAB method. Briefly describe the method and report any modification.
10. Was the DNA quality and concentration checked before proceeding with library prep? If so, please describe it in the methods.
11. Data availability: the sequences are not found in the Genome Sequence Archive. Please make sure that they are publicly accessible.
12. It would be good if the demographics of the healthy control group is reported.
13. In Figure 2D, what do the figure legend "Ratio" mean? Ratio of which variables? Also, the sizes of each node is not distinguishable. Please improve it.
14. In Figure 2C, what do you mean by "NS"?
15. The figure caption for Figure 1D is misleading. The upper panel of Figure 1D means that the metabolites are downregulated in hATTR, but in the figure they seem to be upregulated or at least expressed more in hATTR. Please clarify this.
16. In Figure S2-A, why are there color gradients in between HC and ATTR-v? This seems misleading because it may be interpreted that there is another group in between the two.
17. Please improve the presentation of the figures. Improve the figure layout and be consisted with the order of appearance of the panels (for example, Figure 1A-D are arranged left to right, top bottom).
18. Please improve the grammar and flow.

Reviewer #3 (Comments for the Author):

Li and colleagues investigate the microbiome and metabolome changes associated with the rare disease hereditary transthyretin amyloidosis (hATTR), which is of importance because despite the genetic basis, this condition manifests differently in patients. A modest patient cohort of 13 individuals was assessed in the study in comparison to 22 controls, which is perhaps on the small side given the potential confounding variables that can contribute to disease progression and severity. The authors discovered a decrease in lactic acid bacteria in hATTR individuals, which coincided with a decrease in serum levels of GABA and taurine. The paper's results clearly represented in the figures, and most of the conclusions seem supported by the data with the discussion section being rather lengthy and speculative. I appreciate the authors' discussion of the potential limitations of their study. Below I listed some additional comments and suggestions for the authors to consider:

Line 30 (and throughout): 'probiotics' is only appropriate if these bacteria have been taken as a probiotic supplement. Since these were identified in the microbiota, 'commensals' would be a more appropriate term.

Line 100 and Table 1: Please also provide metrics for the healthy control group, as well as inclusion criteria used for both.

Line 118: It's not clear what these 95 modules are, what are the criteria for grouping, and how this number was achieved.

Given the low n number for the total patient population, could you provide more insight into whether this further subclassification into cardiac amyloidosis, mixed, neurologic and HC is sufficiently statistically powered to yield significant outcomes?

Line 131: Please provide a reference for 2-Furoic Acid as a gut microbial metabolite.

Lines 131-136: I do not understand the rationale for why these metabolites would be of gut microbial as opposed to diet or host origin? If there is no such indication or experimental evidence then this claim is not supported and should be altered.

Line 157: What is a CAG and what value does this analysis bring?

Line 218: "We hypothesize that an increase in *Bifidobacterium scardovii* may correlate with a higher risk of opportunistic

infections in hATTR patients." I don't understand this conclusion, is *B. scardovii* the (opportunistic) infective agent or is it correlated with additional infections? Is this bacterium clinically isolated from hATTR opportunistic infections? If there are other infective agents, then why are these not observed in the microbiota analysis?

Line 230-237: There might be differences in potential SCFA producers, but were there concomitant changes in the SCFA levels from the metabolomics analysis?

Line 233: "Additionally, *Bifidobacterium pseudocatenulatum*, which was found to be depleted in hATTR, has been reported to induce a TLR2-dependent anti-inflammatory response in intestinal lymphocytes of mice with cirrhosis(36)". I'm not sure I understand the relevance of this statement, hATTR and cirrhosis are two different conditions. Also, what is the relevance of the TLR2 stimulation for hATTR specifically?

Line 238-255: There is mixed use of the terms ATTR, hATTR and ATTR-CA. Is this intentionally this specific, and if so is this study sufficiently powered to make these distinctions?

Line 253: "microbiota-associated disruption of glutamate metabolism" The metabolomics study hinted to a potential change in glutamate metabolism, but how is this tied to the microbiota? I might have missed this from the multi-omics analysis but I did not pick up data that support this in the text or Fig. (4)?

Editor in Chief
Microbiology Spectrum
March 17, 2025

Dear Dr. Christina Cuomo and Editorial Team,

Thank you for the opportunity to revise our manuscript titled “**Unraveling Gut Microbiome Alterations and Metabolic Signatures in Hereditary Transthyretin Amyloidosis**” (Spectrum02302-24R1). We appreciate the helpful comments and have tried our best to revise the manuscript accordingly. Our point-by-point responses to the Reviewers’ comments are listed below.

We appreciate the thorough evaluation of our work and hope that our revisions will meet with your approval. Looking forward to hearing from you soon.

Yours sincerely,

Xiaomin Hu

State Key Laboratory of Complex Severe and Rare Diseases, Peking Union Medical College Hospital, Chinese Academy of Medical Sciences & Peking Union Medical College, 100730, Beijing, China.

Shuyang Zhang

Department of Cardiology, State Key Laboratory of Complex Severe and Rare Diseases, Peking Union Medical College Hospital, Chinese Academy of Medical Sciences & Peking Union Medical College, 100730, Beijing, China.

Major points:

Reviewer #1:

1. In Line 39 (Importance), you mentioned that hATTR is influenced also by environmental or host factors. Is diet a factor of this disease? If so, is it also a factor that should be looked into to explore the complex crosstalk between the gut microbiome and disease? Are there previous studies that explored the correlation of diet with the disease?

Response 1: Thank you for your valuable suggestion. Diet has a pivotal role in shaping the composition, function and diversity of the gut microbiome. We fully agree with the importance of exploring the relationship between diet, gut microbiota, and hereditary transthyretin amyloidosis (hATTR). However, since the collection of diet-related data (such as dietary records and nutritional intake surveys) was not included in the initial design of this study, we are currently unable to implement the analysis of diet.

Through literature searches, we have found that no studies have directly reported the association between diet and hATTR. Nevertheless, diet-microbiota-metabolism axis has been studied in neurodegenerative diseases

(such as Alzheimer's disease)(1), metabolic disorders(2), and inflammatory diseases (such as inflammatory bowel disease)(3). hATTR is a multi-systemic disease caused by amyloidosis of atypical transthyretin protein. Diet may influence the disease progression by affecting the gut microbiota and host metabolism. Our study revealed different patterns of gut microbiome and serum metabolome between hATTR patients and healthy controls, which may provide a theoretical basis for future studies combining dietary interventions.

To address your concerns, we have mentioned the limitation in the discussion part and suggested that future studies should systematically integrate dietary data and further explore the impact of dietary factors on hATTR.

The content is as follows (page 12, line 314-317):

“Another limitation is the lack of analysis of dietary factors, given the relatively well-established association between diet and the gut microbiota (4). The integration of dietary data and further exploration of the impact of dietary factors on hATTR are strongly suggested.”

2. In Line 87, the term “ATTRV30M” is not clear. Please define it.

Response 2: Thank you for pointing out this deficiency. ATTRV30M specifically refers to hereditary transthyretin amyloidosis with V30M *TTR* variant which leads to a methionine for valine substitution at residue 30 of the mature TTR protein. We have made revisions and provided explanations in the introduction section.

The content is as follows (page 5, line 86-87):

“ATTRV30M (hATTR with the V30M *TTR* variant, which leads to a methionine for valine substitution at residue 30 of the mature TTR protein)”

3. What was the statistics used to determine the sample size (13 patients with hATTR and 22 healthy controls)? Does this sample size sufficient for gut microbiome analyses?

Response 3: Thank you for your valuable feedback regarding the sample size in our study. We acknowledge that the sample size in our study was not determined through a specific statistical method. This is primarily due to the rarity of hATTR, which has an estimated prevalence of 1-9/100,000(5, 6), and poses significant challenges in sample collection. Consequently, the sample size was determined based on the actual availability of samples.

Determining an appropriate sample size is particularly challenging in clinical microbiome study. According to Casals-Pascual et al.(7), a microbiome study of Crohn's disease recommends approximately 50 samples per group to achieve adequate statistical power for both alpha and beta diversity analyses. However, the

actual sample size should be adjusted based on specific research objectives, study designs and available resources. Given these constraints, our study was designed as an exploratory analysis. To our knowledge, our study represents the first comprehensive integration of gut metagenome and serum metabolome analyses in hATTR. We aim to reveal potential characteristics in the gut microbiome and the metabolome of hATTR patients. While the current sample size is small, it is sufficient to provide valuable preliminary insights.

However, we recognize that the generalizability of our findings may be affected due to the limited sample size. To address this limitation, we have discussed the impact of sample size on the generalizability of our results in the discussion section. We emphasize the need for larger sample sizes in future studies to validate our findings and provide more robust conclusions.

The content is as follows (page 12, line 308-311):

“Given the rarity of hATTR, our study is limited by its limited sample size. This limitation may affect the robustness of the statistical results and the generalizability of our findings. Future studies should aim to validate our conclusions with larger hATTR cohorts.”

4. In the Results, can you clarify the term “modules” used for serum metabolites?

Response 4: Thank you for your feedback. In our study, we conducted hierarchical clustering analysis of the serum metabolome. We first screened for differential metabolites using a combination of statistical criteria: a significance level of $P < 0.05$ (Student's *t*-test) and a Variable Importance in Projection (VIP) score > 1 (OPLS-DA). Subsequently, we applied hierarchical clustering analysis based on these differential metabolites. Co-abundant metabolites were clustered using the Spearman correlation method, which allowed us to identify groups of metabolites with similar abundance patterns. This process resulted in a total of 95 metabolite modules, as we termed them. The cumulative abundance of each module was then calculated by summing the abundances of individual metabolites within the module. The resulting abundance matrix of metabolite modules was used for subsequent multi-omics statistical analyses.

Similar clustering approaches, which are useful for revealing direct correlations between metabolites and discovering novel biomarkers, have been applied in previous metabolomics studies (8-10). In our study, clustering analysis helped us to find that several glutamate derived metabolites (gamma-Glutamylglutamic acid, L-Glutamic acid monosodium salt, Y-Aminobutyric acid) and taurine were grouped into one module (M4). This module was found to be closely associated with disease status through

multi-omics correlation analysis.

To address your concerns, we have revised the Methods section to provide a more detailed explanation of our clustering methodology.

The content is as follows (page 15, line 387-394):

“We then performed a hierarchical clustering analysis of the differential metabolites using the R package *vegan*. Firstly, the distance between differential metabolites was calculated via the Spearman method. Clusters, termed metabolite modules (labelled M1-M95), were subsequently identified via the *t* *hclust* and *cutree* functions with a height threshold of 0.4 and the “average” method. The cumulative abundance of each module was then calculated by summing the abundances of individual metabolites within the module. The resulting abundance matrix of metabolite modules was used for subsequent multiomics statistical analyses.”

Compositions of metabolite modules were listed in Additional file 2: Table S2.

R codes for hierarchical clustering analysis are as followed:

```
setwd("file address")
library(vegan)
library(pheatmap)
library(ggplot2)
library(FactoMineR)
library(factoextra)
library(cluster)
dat <- read.delim('diff_input.txt', row.names = 1, sep = '\t', head = TRUE,
check.names = FALSE)

clust_dist <- get_dist(dat, method = "spearman")
tree <- hclust(clust_dist, method = 'average')
rect.hclust(tree,h=0.4)
label <- cutree(tree,h=0.4)
plot(tree)
head(label)
write.table(label,"file address")
```

The input data, i.e., “diff_input.txt”, is the abundance matrix of differential metabolites. The output file, i.e., “label”, is the result of clustering.

5. The authors mentioned that the controls were age- and sex-matched with the patients. What was the inclusion criterion for age?

Response 5: Thank you for your insightful comments. We appreciate your attention to the details of our study design. Regarding the demographics of our study cohorts, subjects in the healthy control (HC) group ranged in age from 35 to 65 years, with a composition of 11 males and 11 females. In the hATTR group, subjects ranged in age from 33 to 65 years, including 5 males and 8 females. Statistical analysis confirmed that there were no significant differences in age or sex between the two groups.

However, we acknowledge that our description of “age and sex matched” could be misleading, as this was not a strict pairwise matching process. To avoid any potential misunderstanding, we have removed the term “age and sex matched and revised the text to more accurately reflect our study design.

To address your concerns, we have included this detailed demographic information in Additional file 2: Table S1, and revised the Results section.

The contents are as followed (page 5, line 100):

“The study enrolled 13 individuals with hATTR and 22 healthy controls (Table S1).”

Table S1. Baselines of the study population

	HC (N=22)	ATTR-v (N=13)	P-value
Age*	42.87(7.78)	54.00(11.63)	0.13
Male#	11(50.0)	5(38.5)	0.508

*mean \pm SD, #n (%).

Age was compared by Student’s t-test. Sex was compared by χ^2 test. Statistical analyses were performed using SPSS Statistics software, version 27.0 (SPSS Inc., Chicago, IL, USA).

6. I recommend that the authors list down the primers used for the *TTR* gene.

Response 6: Thank you for your valuable suggestion. We have now included the details of the PCR primers in Additional file 2: Table S6, and revised the Methods section.

The content is as follows (page 14, line 355):

“The primers used for amplification are listed in Table S6.”

Table S6. Primers used for amplification of the *TTR* gene

gene	Primer sequence (5' to 3')
Exon2	F: CAATTTTGTTAACTTCTCACG
	R: CAGATGATGTGAGCCTCTCTC

Exon3	F: CCTCCATGCGTAACTTAATCC R: TAGGACATTTCTGTGGTACAC
Exon4	F: TGGTGGAAATGGATCTGTCTG R: TGAAGGGACAATAAGGGAAT

7. I strongly suggest that the authors provide a brief methodology regarding blood and fecal sample collection (volume of blood, weight of fecal samples) and storage.

Response 7: Thank you for your valuable suggestion. To address your concern, we have added the detailed procedures for collecting blood and fecal samples in the Supplementary Methods section.

The content is as follows (Additional file 3: Supplementary Methods, page 1, line 5-13):

“Collection of blood samples

Peripheral fasting blood was drawn in the morning. Peripheral blood samples were centrifuged at 3000 rpm for 5 min after standing at room temperature for at least 30 min, and the supernatant was purified. Serum samples were then stored at – 80 °C immediately.

Collection of stool samples

“Participants were given a stool sampler and provided detailed illustrated instructions for sample collection. Stool samples freshly collected from each participant were immediately transported to the laboratory and frozen at – 80 °C immediately.”

8. Please clarify in the methods that you performed shotgun metagenomics sequencing.

Response 8: Thank you for your kind notification. We have revised the Methods section according to your suggestion to improve clarity.

The content is as follows (page 15, line 396):

“We performed shotgun metagenomics sequencing.”

9. Cite the study protocol used for the CTAB method. Briefly describe the method and report any modification.

Response 9: Thank you for your valuable suggestion. We have revised the Methods section to address your concern. In addition, we have added the details of CTAB method in the Supplementary Methods.

The content is as follows (page 15, line 396-398):

“The extraction of fecal DNA via the CTAB method and library preparation

was accomplished at the Novogene Bioinformatics Technology Co., Ltd (detailed in the Supplementary methods)(11).”

The content is as follows (Additional file 3: Supplementary Methods, page 1-2, line 14-34):

“DNA extraction by CTAB method

1. Pipette 1000 μ L of CTAB lysis buffer into a 2.0 mL EP tube, add lysozyme, and then add an appropriate amount of sample to the lysis buffer. Incubate in a water bath at 65°C, inverting the tube several times during the process to ensure complete lysis of the sample.
2. Centrifuge and collect the supernatant. Add phenol (pH 8.0): chloroform: isoamyl alcohol (25:24:1), invert to mix thoroughly, and centrifuge at 12,000 rpm for 10 minutes.
3. Collect the supernatant again. Add chloroform: isoamyl alcohol (24:1), invert to mix thoroughly, and centrifuge at 12,000 rpm for 10 minutes.
4. Transfer the supernatant to a 1.5 mL centrifuge tube, add isopropanol, and mix by inverting. Precipitate at -20°C.
5. Centrifuge at 12,000 rpm for 10 minutes. Carefully remove the liquid without disturbing the precipitate. Wash the precipitate twice with 1 mL of 75% ethanol. Any remaining liquid can be removed by centrifugation again and then aspirated with a pipette tip.
6. Dry the precipitate in a laminar flow hood or at room temperature (do not over-dry the DNA sample, as it may be difficult to dissolve).
7. Dissolve the DNA sample in ddH₂O. If necessary, incubate at 55-60°C for 10 minutes to aid dissolution.
8. Add 1 μ L of RNase A to digest RNA, and incubate at 37°C for 15 minutes.”

10. Was the DNA quality and concentration checked before proceeding with library prep? If so, please describe it in the methods.

Response 10: Thank you for your suggestion. As you mentioned, the assessment of DNA quality and concentration is an essential step in metagenomic sequencing. We have added the standard procedures for library preparation in the Supplementary Methods section. We hope it will help to avoid any ambiguity regarding our methodology.

The content is as follows (page 15, line 396-398):

“The extraction of fecal DNA via the CTAB method and library preparation was accomplished at the Novogene Bioinformatics Technology Co., Ltd (detailed in the Supplementary methods)(11).”

The content is as follows (Additional file 3: Supplementary Methods, page 2, line 35-46):

“Library preparation

We analyzed the integrity and purity of DNA using agarose gel electrophoresis (AGE) and precisely quantified the DNA concentration using a Qubit fluorometer. For qualified DNA samples, we randomly fragmented them into approximately 350 bp fragments using a Covaris ultrasonicator. Subsequently, the library preparation was completed through a series of steps, including end repair, A-tailing, adapter ligation, purification, and PCR amplification. After the DNA library construction was completed, we performed an initial quantification using Qubit 2.0, diluted the library to 2 ng/μL, and then assessed the insert size of the library using an Agilent 2100 Bioanalyzer. Once the insert size met the expected range, we used quantitative PCR (qPCR) to accurately determine the effective concentration of the library, which must be greater than 3 nM to ensure library quality.”

11. Data availability: the sequences are not found in the Genome Sequence Archive. Please make sure that they are publicly accessible.

Response 11: Thank you for your kind notification. We appreciate your attention to the data availability. The metagenomic sequencing data supporting our study have been deposited in the Genome Sequence Archive (GSA) for Human under controlled access. The data can be accessed through the following link: <https://ngdc.cncb.ac.cn/gsa-human/browse/HRA004297>.

To ensure proper access, we have clarified in the Data Availability section that the data will be made available upon request, following the necessary application procedures. We apologize for any confusion and appreciate your understanding.

The content is as follows (page 17, line 467-470):

“The metagenomic sequencing data supporting the results of this study have been deposited in the Genome Sequence Archive (GSA) for human under BioProject PRJCA015966, with the accession code HRA004297. The data can be available through an online application process.”

12. It would be good if the demographics of the healthy control group is reported.

Response 12: Thank you for your valuable suggestion. We have added Table S1,

which includes information on sex and age, as well as the results of the between-group comparisons.

The content is as follows (Additional file 2: Table S1):

Table S1. Baselines of the study population

	HC (N=22)	ATTR-v (N=13)	P-value
Age*	42.87(7.78)	54.00(11.63)	0.13
Male#	11(50.0)	5(38.5)	0.508

*mean \pm SD, #n (%).

Age was compared by Student's t-test. Sex was compared by χ^2 test. Statistical analyses were performed using SPSS Statistics software, version 27.0 (SPSS Inc., Chicago, IL, USA).

13. In Figure 2D, what do the figure legend "Ratio" mean? Ratio of which variables? Also, the sizes of each node is not distinguishable. Please improve it.

Response 13: Thank you for your kind notification. We apologize for any confusion regarding the figure legends. It appears there was a misunderstanding in the figure reference. We believe the question pertains to Figure 1D, not Figure 2D.

Figure 1D presents the results of metabolite set enrichment analysis (MSEA). We uploaded a list of metabolites that were either hATTR-enriched or hATTR-depleted to the MetaboAnalyst 5.0 platform. This list was cross-referenced against the MetaboAnalyst compound libraries (HMDB, PubChem, KEGG, etc.), resulting in the identification of metabolite sets. The "Ratio" in the figure legend refers to the enrichment ratio of each metabolite set. This score reflects the over-representation of metabolites within a specific set compared to the background metabolome.

For more details on the workflow, please refer to the MetaboAnalyst enrichment analysis documentation available at https://www.metaboanalyst.ca/resources/vignettes/Enrichment_Analysis.html.

To address your concern and avoid any ambiguity, we have clarified the legend of Figure 1D to explicitly state that the "Ratio" represents the enrichment ratio.

To enhance visualization, we resized each node to reflect a scale that is ten times its enrichment ratio, thereby making the nodes more distinguishable. In addition, the variable on the horizontal axis should be "-Log₁₀ (P-value)", and we have already revised it.

The revised Figure 1D and legend are as follows:

(page 23, line 699-704)

“(D) Visualization of metabolite set enrichment analysis (MSEA) based on hATTR-depleted metabolites (left panel) and hATTR-enriched metabolites (right panel). The “Ratio” refers to the enrichment ratio of each metabolite set. Metabolites identified as hATTR-depleted or hATTR-enriched were selected based on the criteria of a Student's t-test P-value < 0.05 and an OPLS-DA VIP value > 1.”

14. In Figure 2C, what do you mean by "NS"?

Response 14: Thank you for your kind notification. It appears there was a misunderstanding in the figure reference. We believe the question pertains to Figure 1D, not Figure 2D. “NS” stands for “not significant”. We have already included a description of this abbreviation in the figure legend to clarify its meaning. We apologize for any confusion this may have caused and thank you for bringing it to our attention.

The content is as follows (page 23, line 696-699):

“(C) Volcano plots demonstrating differential serum metabolites between hATTR and HC groups. Data normalization was achieved by auto-scaling. Students’ t test. P-value < 0.05 is considered statistically significant. NS: not significant.”

15. The figure caption for Figure 1D is misleading. The upper panel of Figure 1D means that the metabolites are downregulated in hATTR, but in the figure they seem to be upregulated or at least expressed more in hATTR. Please clarify this.

Response 15: Thank you for your kind notification. Figure 1D presents the results of metabolite set enrichment analysis (MSEA). Specifically, the left panel displays the MSEA results based on metabolites that were downregulated in hATTR, while the right panel shows the results based on metabolites that were upregulated in hATTR.

To ensure clarity, we have added Table S4-5, which provide details of MSEA results corresponding to Figure 1D. We have also revised the figure caption to

better reflect the content of each panel and avoid any misleading interpretations. We hope these changes will resolve the confusion, and thank you again for your valuable feedback.

The content is as follows (page 23, line 699-704):

“(D) Visualization of metabolite set enrichment analysis (MSEA) based on hATTR-depleted metabolites (left panel) and hATTR-enriched metabolites (right panel). The “Ratio” refers to the enrichment ratio of each metabolite set. Metabolites identified as hATTR-depleted or hATTR-enriched were selected based on the criteria of a Student's t-test P-value < 0.05 and an OPLS-DA VIP value > 1.”

Table S4-5 were as follows:

Table S4. Result of MSEA based on hATTR-enriched metabolites

ID	Total	Expected	Hits	P-value	Enrichment ratio
Citrate cycle (TCA cycle)	20	0.394	2	0.0573	0.10
Phenylalanine, tyrosine and tryptophan biosynthesis	4	0.0788	1	0.0766	0.25
Glyoxylate and dicarboxylate metabolism	32	0.631	2	0.129	0.06
Lipoic acid metabolism	8	0.158	1	0.148	0.13
One carbon pool by folate	9	0.177	1	0.164	0.11
Caffeine metabolism	10	0.197	1	0.181	0.10
Phenylalanine metabolism	10	0.197	1	0.181	0.10
Mucin type O-glycan biosynthesis	12	0.237	1	0.213	0.08
Arginine biosynthesis	14	0.276	1	0.244	0.07
Pentose and glucuronate interconversions	18	0.355	1	0.303	0.06
Fructose and mannose metabolism	20	0.394	1	0.33	0.05
Pentose phosphate pathway	22	0.434	1	0.357	0.05
Purine metabolism	65	1.28	2	0.37	0.03
Glycolysis / Gluconeogenesis	26	0.512	1	0.407	0.04
Alanine, aspartate and glutamate metabolism	28	0.552	1	0.43	0.04

MSEA: metabolite set enrichment analysis

Table S5. Result of MSEA based on hATTR-depleted metabolites

ID	Total	Expected	Hits	P-value	Enrichment ratio
D-Glutamine and D-glutamate metabolism	6	0.158	2	0.00945	0.33
Aminoacyl-tRNA biosynthesis	48	1.26	4	0.0346	0.08
Arginine biosynthesis	14	0.368	2	0.0502	0.14
Histidine metabolism	16	0.42	2	0.0641	0.13
Biosynthesis of unsaturated fatty acids	36	0.946	3	0.066	0.08
Retinol metabolism	17	0.447	2	0.0714	0.12
Pantothenate and CoA biosynthesis	19	0.499	2	0.0869	0.11
Phenylalanine, tyrosine and tryptophan biosynthesis	4	0.105	1	0.101	0.25
Nitrogen metabolism	6	0.158	1	0.148	0.17
Alanine, aspartate and glutamate metabolism	28	0.736	2	0.166	0.07
Glutathione metabolism	28	0.736	2	0.166	0.07
Valine, leucine and isoleucine biosynthesis	8	0.21	1	0.192	0.13
Taurine and hypotaurine metabolism	8	0.21	1	0.192	0.13
Ubiquinone and other terpenoid-quinone biosynthesis	9	0.237	1	0.214	0.11
Glycine, serine and threonine metabolism	33	0.867	2	0.214	0.06

MSEA: metabolite set enrichment analysis

16. In Figure S2-A, why are there color gradients in between HC and ATTR-v? This seems misleading because it may be interpreted that there is another group in between the two.

Response 16: Thank you for your insightful comment. We apologize for any confusion caused by the color gradient in Figure S2A. We understand that it may have been misleading and could be interpreted as indicating an additional group between HC and hATTR. To address your concerns and ensure clarity, we have revised Figure S2A by removing the color gradient.

The revised Figure S2A is as follows:

17. Please improve the presentation of the figures. Improve the figure layout and be consisted with the order of appearance of the panels (for example, Figure 1A-D are arranged left to right, top bottom).

Response 17: Thank you for your valuable suggestion. We have carefully reviewed the layout of the figures and have revised Figure 1 to ensure that the panels are arranged consistently in a left-to-right, top-to-bottom order. We hope these changes address your concern and improve the overall presentation of the figures.

The revised Figure 1 is as follows:

18. Please improve the grammar and flow.

Response 18: Thank you very much for your valuable suggestion. We have carefully revised the manuscript and sought assistance from a professional editing service provided by AJE to further enhance the grammar and flow. We sincerely hope that these improvements will meet your criteria.

The editing certificate has been provided for your reference.

Editing Certificate

This document certifies that the manuscript

Unraveling Gut Microbiome Alterations and Metabolic Signatures in Hereditary Transthyretin Amyloidosis

prepared by the authors

Hanyu Lia, †, Zeyuan Wang, †, Shan Hea, †, Xinyue Zhao, Qingyang Wu, Yueshen Suna, Yue Fana, Xiaomin Hub, #, Zhuang Tiana, #, Shuyang Zhanga, #

was edited for proper English language, grammar, punctuation, spelling, and overall style by one or more of the highly qualified English speaking editors at AJE.

This certificate was issued on **March 13, 2025** and may be verified on the AJE website using the verification code **C77A-21F5-FE9A-6C75-F853**.

Neither the research content nor the authors' intentions were altered in any way during the editing process. Documents receiving this certification should be English-ready for publication; however, the author has the ability to accept or reject our suggestions and changes. To verify the final AJE edited version, please visit our verification page at aje.com/certificate. If you have any questions or concerns about this edited document, please contact AJE at support@aje.com.

AJE provides a range of editing, translation, and manuscript services for researchers and publishers around the world. For more information about our company, services, and partner discounts, please visit aje.com.

Reviewer #2:

1. Line 30 (and throughout): 'probiotics' is only appropriate if these bacteria have been taken as a probiotic supplement. Since these were identified in the microbiota, 'commensals' would be a more appropriate term.

Response 1: Thank you very much for your valuable advice, which has greatly helped us achieve more accurate expression. We have replaced the term “probiotics” with “commensals” throughout the manuscript following your suggestion.

For example, the revised Line 30 (page 2) is as follows:

“Additionally, commensals such as *Bifidobacterium pseudocatenulatum*, *Lactobacillus rogosae*, and *Hungatella hathewayi* were significantly diminished in hATTR patients and were positively correlated with the metabolite module containing GABA and taurine.”

2. Line 100 and Table 1: Please also provide metrics for the healthy control group, as well as inclusion criteria used for both.

Response 2: Thank you very much for your valuable suggestion, which has been extremely helpful in enhancing the completeness of our work. We have now included Table S1, which provides detailed metrics for the healthy control group, including information on sex and age, as well as the results of between-group

comparisons.

The content is as follows (Additional file 2: Table S1):

Table S1. Baselines of the study population

	HC (N=22)	ATTR-v (N=13)	P-value
Age*	42.87(7.78)	54.00(11.63)	0.13
Male#	11(50.0)	5(38.5)	0.508

*mean \pm SD, #n (%).

Age was compared by Student's *t*-test. Sex was compared by χ^2 test. Statistical analyses were performed using SPSS Statistics software, version 27.0 (SPSS Inc., Chicago, IL, USA).

Additionally, the inclusion criteria for both hATTR individuals and healthy controls have already been described in the Methods section (page 13, line 330-351). We hope this clarifies the information and apologize for any previous ambiguity.

3. Line 118: It's not clear what these 95 modules are, what are the criteria for grouping, and how this number was achieved.

Response 3: Thank you very much for your insightful question. In our study, we conducted hierarchical clustering analysis of the serum metabolome. We first screened for differential metabolites using a combination of statistical criteria: a significance level of $P < 0.05$ (Student's *t*-test) and a Variable Importance in Projection (VIP) score > 1 (OPLS-DA). Subsequently, we applied hierarchical clustering analysis based on these differential metabolites. Co-abundant metabolites were clustered using the Spearman correlation method, which allowed us to identify groups of metabolites with similar abundance patterns. This process resulted in a total of 95 metabolite modules, and compositions of metabolite modules are listed in Additional file 2: Table S2.

Similar clustering approaches, which are useful for revealing direct correlations between metabolites and discovering novel biomarkers, have been applied in previous metabolomics studies (8-10). In our study, clustering analysis helped us to identified that several glutamate derived metabolites (e.g., gamma-Glutamylglutamic acid, L-Glutamic acid monosodium salt, Y-Aminobutyric acid) and taurine were grouped into one module (M4). This module was found to be closely associated with disease status through multi-omics correlation analysis.

To address your concerns, we have revised the Methods section to provide a more detailed explanation of our clustering methodology. We hope this clarifies

the process and rationale behind the identification of the 95 metabolite modules. Thank you again for your valuable feedback.

The content is as follows (page 15, line 387-394):

“We then performed a hierarchical clustering analysis of the differential metabolites using the R package `vegan`. Firstly, the distance between differential metabolites was calculated via the Spearman method. Clusters, termed metabolite modules (labelled M1-M95), were subsequently identified via the `t hclust` and `cutree` functions with a height threshold of 0.4 and the “average” method. The cumulative abundance of each module was then calculated by summing the abundances of individual metabolites within the module. The resulting abundance matrix of metabolite modules was used for subsequent multiomics statistical analyses.”

```
setwd("file address")
library(vegan)
library(pheatmap)
library(ggplot2)
library(FactoMineR)
library(factoextra)
library(cluster)

dat <- read.delim('diff_input.txt', row.names = 1, sep = '\t', head = TRUE,
check.names = FALSE)

clust_dist <- get_dist(dat, method = "spearman")
tree <- hclust(clust_dist, method = 'average')
rect.hclust(tree,h=0.4)
label <- cutree(tree,h=0.4)
plot(tree)
head(label)
write.table(label,"file address")
```

R codes for hierarchical clustering analysis are as followed:

The input data, i.e., “diff_input.txt”, is the abundance matrix of differential metabolites. The output file, i.e., “label”, is the result of clustering

4. Given the low n number for the total patient population, could you provide more

insight into whether this further subclassification into cardiac amyloidosis, mixed, neurologic and HC is sufficiently statistically powered to yield significant outcomes?

Response 4: We sincerely appreciate your insightful question regarding the statistical power of subgroup analyses.

We fully acknowledge the limited sample size in the subgroups, including Carrier (N=2), Neurologic (N=5), and Mix (both neurologic phenotype and evidence of cardiac amyloidosis, N=6), as shown in Table 1, which is an inherent challenge in studying hereditary ATTR amyloidosis—a rare disease with an estimated prevalence of 1-9/100,000(5, 6).

Despite this limitation, we intentionally pursued subgroup analyses due to the clinical importance of identifying phenotype-specific biomarkers. Hereditary ATTR amyloidosis (hATTR) is clinically heterogeneous, and the prognosis is highly associated with disease phenotypes. For example, cardiac amyloidosis is a particularly concerning sign that points to a significantly poorer prognosis (12, 13). Therefore, identifying microbiome/metabolite signatures specific to these phenotypes could inspire future mechanistic studies or personalized intervention tools.

In our study, we found that the metabolite module 4 (M4), consisting of glutamate derivatives and taurine, was downregulated in hATTR and negatively correlated with the cardiac amyloidosis phenotype. This phenomenon could provide direction for further mechanistic exploration.

As shown in Figure 2B and 4C, comparisons of certain metabolite or microbial species between subgroups were performed by Wilcoxon rank sum test, with $P < 0.05$ considered statistically significant. We recognize that the small sample size may impact the robustness of our results and increase the risk of overinterpretation. However, we believe that our findings still offer valuable clinical insights, although they require further validation through additional independent studies.

To address your concerns, we have clarified the limitations regarding the sample size in the Discussion section.

The content is as follows (page 12, line 308-311):

“Given the rarity of hATTR, our study is limited by its limited sample size. This limitation may affect the robustness of the statistical results and the generalizability of our findings. Future studies should aim to validate our conclusions with larger hATTR cohorts.”

5. Line 131: Please provide a reference for 2-Furoic Acid as a gut microbial

metabolite.

Response 5: Thank you for your kind notification. We have added a reference to support the statement that 2-Furoic acid could be derived from gut microbiota (PMID: 22031465).

The revised content is as follows (page 6, line 130-132):

“Notably, compared with other metabolites in M1, 2-furoic acid (2-FA), potentially derived from the gut microbiota(14), showed relatively superior diagnostic potential, with an AUC of 0.897.”

According to this review, several microorganisms, mostly Gram-negative aerobic bacteria (as shown in Table 1 of the review), are known to degrade furanic compounds. 2-Furoic Acid is an intermediate metabolite produced during this degradation process. This finding highlights the role of gut microbiota in the metabolism of furanic compounds and the production of related metabolites.

6. Lines 131-136: I do not understand the rationale for why these metabolites would be of gut microbial as opposed to diet or host origin? If there is no such indication or experimental evidence then this claim is not supported and should be altered.

Response 6: Thank you for your valuable suggestion. In our manuscript, we aimed to highlight the potential association of these metabolites with the gut microbiota. As described in Response 5, we have already added a reference which suggested that 2-Furoic acid might be of microbial origin. As for what we mentioned in Lines 135-135, aspartic acid and GABA can be produced by gut microbiota, as evidenced by previous studies (PMID: 38806059 and PMID: 30531975) (15, 16). Additionally, taurine serves as an important metabolic substrate for gut microbiota(17).

We acknowledge that the previous expression may not have been sufficiently accurate. Following your suggestion, we have revised the language and added corresponding references to enhance its precision and clarity.

The content is as follows (page 6, line 132-135):

“Additionally, module M4, composed primarily of several amino acids and their derivatives (such as aspartic acid, GABA, and taurine) which may be closely associated with the gut microbiota(15-17),, exhibited a strong classification effect between the hATTR and HC groups, with an AUC of 0.941.”

7. Line 157: What is a CAG and what value does this analysis bring?

Response 7: Thank you for your valuable question. In our study, we performed a guild-based analysis to identify co-abundant species that may function as coherent functional groups within the gut ecosystem. The term “guild” originates from

ecological studies of macro-organisms, where members are defined as belonging to the same guild if they exploit the same class of resources in a similar way or work together as a functional unit. When applied to gut microbiota research, a guild comprises microbial species that exhibit co-abundance patterns, thriving or declining together regardless of their taxonomic classifications whenever resources become available or depleted. This approach is well-documented in the literature (e.g., PMID: 33563315) (18).

In our context, a CAG (Co-Abundant Group) refers to a cluster of microbial species identified through hierarchical clustering based on their co-abundance patterns. The primary advantage of this analysis strategy is to reduce dimensionality and sparsity in microbiome-wide association studies. This allows us to identify candidate gut bacteria that may contribute to human health and disease. Similar analytical strategies have been employed in previous studies, including a publication from our research team(8) and a notable work by Liping Zhao et al (19).

To address your concerns and avoid potential ambiguity, we have revised the relevant sections to provide a clearer explanation of the guild concept. Additionally, we have detailed the procedures for constructing the species interaction network and acquiring CAGs in the Methods section.

The content is as follows (page 7, line 156-157):

“We then performed a guild-based analysis to identify co-abundant species that might work as functional groups in the gut ecosystem(18, 19). To achieve this, we calculated the Spearman correlations between the 106 key species and constructed a co-occurrence network (Figure 3E). This analysis yielded a total of 46 co-abundant groups (CAGs) (Table S3).”

The revised Methods section is as follows (page 17, line 442-450):

“A hierarchical clustering analysis of the key species was then performed using the R package *vegan*. The distances between key species were computed using the Spearman method via the *vegan* package. The co-occurrence network of key species was visualized using Cytoscape v3.8.0. Then, clusters, termed co-abundance groups (CAGs, labelled CAG1-46), were identified via the *hclust* and *cutree* functions with a height threshold of 0.5 and the “average” method. The cumulative abundance of each CAG was then calculated by summing the abundances of individual species within the group. The resulting abundance matrix of CAGs was used for subsequent multi-omics statistical analyses.”

8. Line 218: "We hypothesize that an increase in *Bifidobacterium scardovii* may correlate with a higher risk of opportunistic infections in hATTR patients." I don't

understand this conclusion, is *B. scardovii* the (opportunistic) infective agent or is it correlated with additional infections? Is this bacterium clinically isolated from hATTR opportunistic infections? If there are other infective agents, then why are these not observed in the microbiota analysis?

Response 8: Thank you for your insightful comment. In our study, *Bifidobacterium scardovii* was identified as a key differential microbe between the HC and hATTR groups, exhibiting the highest VIP value through PLS-DA analysis. There are two reported cases of *Bifidobacterium scardovii* causing infections: one involving an elderly male patient with breast abscess, and the other involving an elderly female patient with a history of cancer and chemotherapy who developed recurrent urinary tract infection (20, 21).

In the context of hATTR, patients usually experience debilitation and compromised immune function. Therefore, we hypothesized that the enrichment of *Bifidobacterium scardovii* of hATTR patients might be a marker of gut microbiota imbalance and could be associated with a higher risk of opportunistic infections. However, we acknowledge that this hypothesis may be overinterpreted without direct clinical evidence linking *B. scardovii* to specific infections in hATTR patients.

To address your concerns, we have revised the discussion section to clarify the potential pathogenic role of *Bifidobacterium scardovii*. We hope this revision provides a more balanced and plausible explanation.

The content is as follows (page 9, line 219-227):

“Among the microbes enriched in hATTR patients, *Bifidobacterium scardovii* exhibited the highest VIP-value through PLS-DA analysis, as shown in Figure 3D. It is a slow-growing, non-spore-forming anaerobic gram-positive bacillus. There are two reported cases of *Bifidobacterium scardovii* causing infections: one involving an elderly male patient with breast abscess, and the other involving an elderly female patient with a history of cancer and chemotherapy who developed recurrent urinary tract infection (20, 21). Given the frailty of hATTR patients, whether the enrichment of *Bifidobacterium scardovii* is associated with a more fragile gut condition and a higher risk of infection requires further exploration.”

9. Line 230-237: There might be differences in potential SCFA producers, but were there concomitant changes in the SCFA levels from the metabolomics analysis?

Response 9: Thank you for your insightful question. In our analysis of the commensal microbes depleted in hATTR, we identified several species with the potential to produce short-chain fatty acids (SCFAs). However, due to the limitations of our untargeted metabolomics approach, we were unable to

systematically detect and quantify SCFAs (e.g., acetate, propionate and butyrate) in either serum or fecal samples.

To address your concerns, we have revised the discussion section to highlight this limitation and outline future directions. Specifically, we have emphasized the need for targeted metabolomics to accurately measure SCFA levels, as well as the necessity for mechanistic studies to elucidate the relationship between the intestinal barrier and SCFA-producing microbes in hATTR.

The content is as follows (page 10, line 233-243):

“Regarding the hATTR-depleted microbes, CAG3 (featuring *Bifidobacterium pseudocatenulatum*) and CAG4 (comprising *Eubacterium* species and *Lactobacillus rogosae*) were both found to be significantly downregulated in hATTR patients. It is important to note that *Bifidobacterium*, *Eubacterium* and *Lactobacillus* are prominent genera known for their production of short-chain fatty acids (SCFAs)(22, 23) SCFAs, particularly butyrate, are essential for maintaining intestinal barrier function and modulating mucosal immunity(24, 25). Whether the reduction of the above commensals leads to decreased levels of serum and fecal SCFAs needs to be further explored through targeted metabolomics. Additionally, whether the reduction of these commensals contributes to the disruption of the intestinal barrier in hATTR patients remains uncertain and requires further mechanistic studies.”

10. Line 233: "Additionally, *Bifidobacterium pseudocatenulatum*, which was found to be depleted in hATTR, has been reported to induce a TLR2-dependent anti-inflammatory response in intestinal lymphocytes of mice with cirrhosis(36)". I'm not sure I understand the relevance of this statement, hATTR and cirrhosis are two different conditions. Also, what is the relevance of the TLR2 stimulation for hATTR specifically?

Response 10: Thank you for your valuable comment. We intended to discuss the potential anti-inflammatory role of the hATTR depleted *Bifidobacterium pseudocatenulatum*, and to explore whether its downregulation might influence the inflammatory status in hATTR patients. However, as you correctly pointed out, hATTR and cirrhosis are distinct conditions, and the reference to cirrhosis in our previous discussion was not directly relevant to hATTR. This may have led to over-interpretation and introduced unnecessary complexity to the discussion.

After careful consideration, we have removed this part from the manuscript to ensure that our discussion better align our conclusions with the data presented.

11. Line 238-255: There is mixed use of the terms ATTR, hATTR and ATTR-CA. Is

this intentionally this specific, and if so is this study sufficiently powered to make these distinctions?

Response 11: Thank you for your kind feedback. Transthyretin amyloidosis (ATTR) can be classified into two main types: wild-type ATTR and hereditary ATTR (hATTR). In our study, we specifically focused on hereditary transthyretin amyloidosis (hATTR). The term ATTR-CA refers to cardiac amyloidosis that is secondary to ATTR. In our study, six of the hATTR patients exhibited cardiac amyloidosis, as detailed in Table 1.

To address your concerns, we have carefully reviewed all expressions related to hATTR throughout the manuscript to ensure consistency and clarity. Additionally, we have replaced the term ATTR-CA with “cardiac amyloidosis” to avoid any potential misunderstandings.

12. Line 253: "microbiota-associated disruption of glutamate metabolism" The metabolomics study hinted to a potential change in glutamate metabolism, but how is this tied to the microbiota? I might have missed this from the multi-omics analysis but I did not pick up data that support this in the text or Fig. (4)?

Response 13: Thank you for your valuable suggestion. In our study, the results suggest abnormalities in glutamate metabolism from both the metabolome and metagenome perspectives. Specifically:

- 1) Metabolome analysis: The metabolite module M4, which consists of various glutamate derivatives, was significantly downregulated in hATTR. Metabolite set enrichment analysis (MSEA) also revealed that metabolites depleted in hATTR were enriched in glutamate metabolism pathways, as shown in Figure 1D (left panel).
- 2) Metagenomic Analysis: Metagenomic pathway analysis indicated that KEGG orthologs (KOs) downregulated in hATTR were enriched in pathways related to glutamate metabolism (Figure 5C). As shown in Figure 4B, Co-Abundant Groups (CAGs) that include potential GABA-producing microbes (*Bifidobacterium* spp. and *Lactobacillus* spp., referred to (26).) were positively correlated with module M4.

However, as you pointed out, the direct association between microbiota and glutamate metabolism remains ambiguous. Our previous discussion may have been overly interpretative without sufficient evidence to support a clear mechanistic link.

To address your concerns, we have revised the discussion section to make it more logical and better aligned with our results. We have also revised the conclusion section to avoid over-interpreting the data.

The revised discussion section is as follows (page 10, line 244-266):

“An integrated analysis of the microbiome and metabolome revealed that multiple CAGs depleted in hATTR were positively correlated with metabolite module M4, which consists of various glutamate derivatives (Figure 4B). M4 was downregulated in hATTR and negatively correlated with cardiac amyloidosis. Furthermore, metagenomic pathway analysis indicated that KEGG orthologs (KOs) downregulated in hATTR were enriched in pathways related to glutamate metabolism (Figure 5C, left panel). Collectively, these findings suggest abnormalities in glutamate metabolism from both the metabolome and metagenome perspectives. Specifically, we observed downregulation of gamma-aminobutyric acid (GABA) in the hATTR group. GABA is a crucial inhibitory neurotransmitter that plays a role in numerous physiological and psychological processes. Previous studies have shown that GABA can be produced by various gut bacterial genera, such as *Bifidobacterium*, *Lactobacillus* and *Bacteroides* (15, 26). Individuals with ATTR have been shown to be particularly susceptible to psychological distress and psychiatric disorders(27, 28). The relationship between this vulnerability and the dysregulation of glutamate metabolism warrants further investigation. GABA, which mediates the gut-brain axis, also has immunoregulatory functions, promoting the differentiation of monocytes into anti-inflammatory macrophages that secrete interleukin-10 and suppress CD8+ T cell cytotoxicity(29). In our study, the results indicate that abnormal glutamate metabolism appears to be a distinctive feature of hATTR. The specific roles of GABA and other glutamate metabolites in this context require further exploration. Additionally, whether gut dysbiosis contributes to the disruption of glutamate metabolism in hATTR warrants further investigation.”

The revised conclusion section is as follows (page 13, line 320-327):

“In conclusion, our study revealed a distinct pattern of gut dysbiosis in hATTR, characterized by a reduction in commensals, such as *Bifidobacterium pseudocatenulatum*, *Lactobacillus rogosae*, and *Hungatella hathewayi*, and an increase in symbionts or opportunistic pathogens, such as *Parabacteroides* spp.. Additionally, we've identified perturbations in the metabolism of glutamate and taurine in hATTR. Gamma-aminobutyric acid (GABA) and taurine may serve as key metabolites that bridge the gut microbiota with the pathophysiology of hATTR, warranting further investigation.”

References

1. Dissanayaka DMS, Jayasena V, Rainey-Smith SR, Martins RN, Fernando

- WMADB. 2024. The Role of Diet and Gut Microbiota in Alzheimer's Disease. *Nutrients* 16:412.
2. Moszak M, Szulińska M, Bogdański P. 2020. You Are What You Eat—The Relationship between Diet, Microbiota, and Metabolic Disorders—A Review. *Nutrients* 12:1096.
 3. Sugihara K, Kamada N. 2021. Diet–Microbiota Interactions in Inflammatory Bowel Disease. *Nutrients* 13:1533.
 4. Ross FC, Patangia D, Grimaud G, Lavelle A, Dempsey EM, Ross RP, Stanton C. 2024. The interplay between diet and the gut microbiome: implications for health and disease. *Nat Rev Microbiol* 22:671-686.
 5. Adams D, Koike H, Slama M, Coelho T. 2019. Hereditary transthyretin amyloidosis: a model of medical progress for a fatal disease. *Nat Rev Neurol* 15:387-404.
 6. Luigetti M, Romano A, Di Paolantonio A, Bisogni G, Sabatelli M. 2020. Diagnosis and Treatment of Hereditary Transthyretin Amyloidosis (hATTR) Polyneuropathy: Current Perspectives on Improving Patient Care. *Ther Clin Risk Manag* 16:109-123.
 7. Casals-Pascual C, González A, Vázquez-Baeza Y, Song SJ, Jiang L, Knight R. 2020. Microbial Diversity in Clinical Microbiome Studies: Sample Size and Statistical Power Considerations. *Gastroenterology* 158:1524-1528.
 8. Liu H, Chen X, Hu X, Niu H, Tian R, Wang H, Pang H, Jiang L, Qiu B, Chen X, Zhang Y, Ma Y, Tang S, Li H, Feng S, Zhang S, Zhang C. 2019. Alterations in

the gut microbiome and metabolism with coronary artery disease severity.

Microbiome 7:68.

9. Shen B, Yi X, Sun Y, Bi X, Du J, Zhang C, Quan S, Zhang F, Sun R, Qian L, Ge W, Liu W, Liang S, Chen H, Zhang Y, Li J, Xu J, He Z, Chen B, Wang J, Yan H, Zheng Y, Wang D, Zhu J, Kong Z, Kang Z, Liang X, Ding X, Ruan G, Xiang N, Cai X, Gao H, Li L, Li S, Xiao Q, Lu T, Zhu Y, Liu H, Chen H, Guo T. 2020. Proteomic and Metabolomic Characterization of COVID-19 Patient Sera. *Cell* 182:59-72.e15.
10. Hakimi AA, Reznik E, Lee CH, Creighton CJ, Brannon AR, Luna A, Aksoy BA, Liu EM, Shen R, Lee W, Chen Y, Stirdivant SM, Russo P, Chen YB, Tickoo SK, Reuter VE, Cheng EH, Sander C, Hsieh JJ. 2016. An Integrated Metabolic Atlas of Clear Cell Renal Cell Carcinoma. *Cancer Cell* 29:104-116.
11. Minas K, McEwan NR, Newbold CJ, Scott KP. 2011. Optimization of a high-throughput CTAB-based protocol for the extraction of qPCR-grade DNA from rumen fluid, plant and bacterial pure cultures. *FEMS Microbiology Letters* 325:162-169.
12. Castiglione V, Franzini M, Aimo A, Carecci A, Lombardi CM, Passino C, Rapezzi C, Emdin M, Vergaro G. 2021. Use of biomarkers to diagnose and manage cardiac amyloidosis. *Eur J Heart Fail* 23:217-230.
13. Adams D, Algalarrondo V, Polydefkis M, Sarswat N, Slama MS, Nativi-Nicolau J. 2021. Expert opinion on monitoring symptomatic hereditary transthyretin-mediated amyloidosis and assessment of disease progression.

- Orphanet Journal of Rare Diseases 16:411.
14. Wierckx N, Koopman F, Ruijsenaars HJ, de Winde JH. 2011. Microbial degradation of furanic compounds: biochemistry, genetics, and impact. *Applied Microbiology and Biotechnology* 92:1095-1105.
 15. Strandwitz P, Kim KH, Terekhova D, Liu JK, Sharma A, Levering J, McDonald D, Dietrich D, Ramadhar TR, Lekbua A, Mroue N, Liston C, Stewart EJ, Dubin MJ, Zengler K, Knight R, Gilbert JA, Clardy J, Lewis K. 2019. GABA-modulating bacteria of the human gut microbiota. *Nature Microbiology* 4:396-403.
 16. Yoo W, Shealy NG, Zieba JK, Torres TP, Baltagulov M, Thomas JD, Shelton CD, McGovern AG, Foegeding NJ, Olsan EE, Byndloss MX. 2024. Salmonella Typhimurium expansion in the inflamed murine gut is dependent on aspartate derived from ROS-mediated microbiota lysis. *Cell Host & Microbe* 32:887-899.e6.
 17. Qian W, Li M, Yu L, Tian F, Zhao J, Zhai Q. 2023. Effects of Taurine on Gut Microbiota Homeostasis: An Evaluation Based on Two Models of Gut Dysbiosis. *Biomedicines* 11.
 18. Wu G, Zhao N, Zhang C, Lam YY, Zhao L. 2021. Guild-based analysis for understanding gut microbiome in human health and diseases. *Genome Medicine* 13:22.
 19. Zhao L, Zhang F, Ding X, Wu G, Lam YY, Wang X, Fu H, Xue X, Lu C, Ma J, Yu L, Xu C, Ren Z, Xu Y, Xu S, Shen H, Zhu X, Shi Y, Shen Q, Dong W, Liu R,

- Ling Y, Zeng Y, Wang X, Zhang Q, Wang J, Wang L, Wu Y, Zeng B, Wei H, Zhang M, Peng Y, Zhang C. 2018. Gut bacteria selectively promoted by dietary fibers alleviate type 2 diabetes. *Science* 359:1151.
20. Sunil Krishna M, Shenoy PA, Priyanka KS, Vishwanath S. 2022. "Breast abscess in a male patient due to *Fingoldia magna* and *Bifidobacterium scardovii*: An unusual entity". *Anaerobe* 75:102536.
21. Barberis CM, Cittadini RM, Almuzara MN, Feinsilberg A, Famiglietti AM, Ramírez MS, Vay CA. 2012. Recurrent urinary infection with *Bifidobacterium scardovii*. *J Clin Microbiol* 50:1086-8.
22. Lee J, d'Aigle J, Atadja L, Quaiocoe V, Honarpisheh P, Ganesh BP, Hassan A, Graf J, Petrosino J, Putluri N, Zhu L, Durgan DJ, Bryan RM, Jr., McCullough LD, Venna VR. 2020. Gut Microbiota-Derived Short-Chain Fatty Acids Promote Poststroke Recovery in Aged Mice. *Circ Res* 127:453-465.
23. Sun M, Wu W, Liu Z, Cong Y. 2017. Microbiota metabolite short chain fatty acids, GPCR, and inflammatory bowel diseases. *J Gastroenterol* 52:1-8.
24. Mathewson ND, Jenq R, Mathew AV, Koenigskecht M, Hanash A, Toubai T, Oravec-Wilson K, Wu S-R, Sun Y, Rossi C, Fujiwara H, Byun J, Shono Y, Lindemans C, Calafiore M, Schmidt TM, Honda K, Young VB, Pennathur S, van den Brink M, Reddy P. 2016. Gut microbiome-derived metabolites modulate intestinal epithelial cell damage and mitigate graft-versus-host disease. *Nature Immunology* 17:505-513.
25. Marizzoni M, Cattaneo A, Mirabelli P, Festari C, Lopizzo N, Nicolosi V,

- Mombelli E, Mazzelli M, Luongo D, Naviglio D, Coppola L, Salvatore M, Frisoni GB. 2020. Short-Chain Fatty Acids and Lipopolysaccharide as Mediators Between Gut Dysbiosis and Amyloid Pathology in Alzheimer's Disease. *Journal of Alzheimer's Disease* : JAD 78:683-697.
26. Yunes RA, Poluektova EU, Dyachkova MS, Klimina KM, Kovtun AS, Averina OV, Orlova VS, Danilenko VN. 2016. GABA production and structure of gadB/gadC genes in *Lactobacillus* and *Bifidobacterium* strains from human microbiota. *Anaerobe* 42:197-204.
27. Lopes A, Fonseca I, Sousa A, Rodrigues C, Branco M, Coelho T, Sequeiros J, Freitas P. 2018. Psychopathological dimensions in subjects with hereditary ATTR V30M amyloidosis and their relation with life events due to the disease. *Amyloid* 25:26-36.
28. Smorti M, Ponti L, Soffio F, Argirò A, Perfetto F, Zampieri M, Mazzoni C, Tomberli A, Allinovi M, Di Mario C, Olivotto I, Cappelli F. 2022. Prevalence of anxiety and depression symptoms in a sample of outpatients with ATTR cardiac amyloidosis. *Front Psychol* 13:1066224.
29. Zhang B, Vogelzang A, Miyajima M, Sugiura Y, Wu Y, Chamoto K, Nakano R, Hatae R, Menzies RJ, Sonomura K, Hojo N, Ogawa T, Kobayashi W, Tsutsui Y, Yamamoto S, Maruya M, Narushima S, Suzuki K, Sugiya H, Murakami K, Hashimoto M, Ueno H, Kobayashi T, Ito K, Hirano T, Shiroguchi K, Matsuda F, Suematsu M, Honjo T, Fagarasan S. 2021. B cell-derived GABA elicits IL-10(+) macrophages to limit anti-tumour immunity. *Nature* 599:471-476.

Re: Spectrum02302-24R1 (Unraveling Gut Microbiome Alterations and Metabolic Signatures in Hereditary Transthyretin Amyloidosis)

Dear Dr. Xiaomin Hu:

Thank you for carefully addressing the Reviewers' comments. I hereby would like to congratulate you on the acceptance of your manuscript for publication in Spectrum!

Your manuscript has been accepted, and I am forwarding it to the ASM production staff for publication. Your paper will first be checked to make sure all elements meet the technical requirements. ASM staff will contact you if anything needs to be revised before copyediting and production can begin. Otherwise, you will be notified when your proofs are ready to be viewed.

Sincerely,
Jan Claesen
Editor
Microbiology Spectrum